# Operator Splitting with Hamilton-Jacobi-based Proximals

**Nicholas Di** [1]  **Eric C. Chi** [2]  **Samy Wu Fung** [3]

## Abstract

Operator splitting algorithms are a cornerstone of modern first-order optimization, decomposing complex problems into simpler subproblems solved via proximal operators. However, most functions lack closed-form proximal operators, which has long restricted these methods to a narrow set of problems. Hamilton-Jacobi-based proximal operator (HJ-Prox) is a recent derivative-free Monte Carlo technique based on Hamilton-Jacobi PDE theory, that approximates proximal operators numerically. In this work, we introduce a unified framework for operator splitting via HJ-Prox, which allows for deployment of operator splitting even when functions are not proximable. We prove that replacing exact proximal steps with HJ-Prox in algorithms such as proximal point, proximal gradient descent, Douglas–Rachford splitting, Davis–Yin splitting, and primal–dual hybrid gradient preserves convergence guarantees under mild assumptions. Numerical experiments demonstrate HJ-Prox is competitive and effective on a wide variety of statistical learning tasks.

## 1. Introduction

Splitting algorithms are central to modern statistical machine learning and optimization, particularly for problems with nonsmooth composite objectives (Parikh & Boyd, 2014). These methods work by decomposing difficult problems into sequences of simpler subproblems, each involving a proximal operator. The main computational bottleneck arises when proximal operators lack closed-form solutions, forcing practitioners to solve expensive inner optimization problems at each iteration (Tibshirani, 2017; Tibshirani & Taylor, 2011). Recent work (Osher et al., 2023) introduced

---

[1]Department of Statistics, Rice University, Houston TX, USA [2]Department of Statistics, University of Minnesota, Minneapolis MN, USA [3]Department of Applied Mathematics and Statistics, Colorado School of Mines, Golden CO, USA. Correspondence to: Nicholas Di <nd56@rice.edu>.

*Proceedings of the 43rd International Conference on Machine Learning*, Seoul, South Korea. PMLR 306, 2026. Copyright 2026 by the author(s).

HJ-Prox, a Monte Carlo scheme that approximates proximal operators using Hamilton–Jacobi partial differential equations. This approach sidesteps the need for closed-form expressions, but its theoretical properties within splitting frameworks have remained unexplored. We address this gap by establishing convergence guarantees for HJ-Prox when embedded in standard splitting architectures.

### 1.1. Contributions

Our contributions can be summarized as follows.

- We develop a unified convergence theory for zeroth-order splitting algorithms built on HJ-Prox approximations. Our framework applies to four major splitting schemes: proximal gradient descent (PGD), Douglas-Rachford splitting (DRS), Davis-Yin splitting (DYS), and primal-dual hybrid gradient (PDHG) (Ryu & Yin, 2022).
- For each HJ-Prox-based splitting algorithm, we prove convergence almost surely under mild regularity conditions.
- Numerical experiments on sparse regression, trend filtering, and image denoising problems demonstrate that HJ-based splitting methods match the performance of analytically-derived solutions while extending to settings where *closed-form proximal operators are unavailable*.

## 2. Background

Splitting algorithms are designed to solve composite convex optimization problems of the form

$$\min_x f(x) + g(x), \tag{1}$$

where $f$ and $g$ are proper, lower-semicontinuous (LSC) and convex. Their efficiency, however, depends critically on the availability of closed-form proximal operators for $f$ or $g$. When these operators are unavailable, the proximal step must be approximated through iterative subroutines, creating a substantial computational burden or the problem must be reformulated. To address this challenge, several lines of research have emerged. One approach focuses on improving efficiency through randomization within the algorithmic structure. These methods reduce computational cost by sampling blocks of variables, probabilistically skipping the proximal step, or solving suboptimization problems incompletely with controlled error (Mishchenko et al.,

2022; Bonettini et al., 2020; Briceño-Arias et al., 2019; Condat & Richtárik, 2022). Alternatively, other approaches reformulate the problem by focusing on dual formulations (Tibshirani, 2017; Mazumder & Hastie, 2012).

While these techniques improve efficiency, they all share common limitations. They require fluency in proximal calculus to derive the proximal operator. Moreover, proximal operators remain problem-dependent as they typically require tailored solution strategies for each specific function class. This creates a critical research gap: the need for a simple generalizable method that can approximate the proximal operator for a more general class of functions.

### 2.1. Hamilton-Jacobi-based Proximal (HJ-Prox)

A promising solution to this challenge has emerged from recent work that approximates the proximal operator using a Monte Carlo approach inspired by Hamilton-Jacobi (HJ) PDEs. Notably, (Osher et al., 2023) showed that for $\delta > 0$,

$$\text{prox}_{tf}(x) = \lim_{\delta \to 0^+} \frac{\mathbb{E}_{y \sim \mathcal{N}(x, \delta t I)} \left[ y \cdot \exp(-f(y)/\delta) \right]}{\mathbb{E}_{y \sim \mathcal{N}(x, \delta t I)} \left[ \exp(-f(y)/\delta) \right]} \quad (2)$$

$$\approx \frac{\mathbb{E}_{y \sim \mathcal{N}(x, \delta t I)} \left[ y \cdot \exp(-f(y)/\delta) \right]}{\mathbb{E}_{y \sim \mathcal{N}(x, \delta t I)} \left[ \exp(-f(y)/\delta) \right]} \quad (3)$$

$$= \text{prox}_{tf}^{\delta}(x) \quad (4)$$

where $\mathcal{N}(x, \delta t I)$ represents the normal distribution with mean $x$ and covariance matrix $\delta t I$, $t > 0$, and $f$ is assumed to be weakly-convex (Ryu & Yin, 2022). The HJ-Prox, denoted by $\text{prox}_{tf}^{\delta}$ in (4), fixes a small value of $\delta > 0$ to approximate the limiting expression above, which enables a Monte Carlo approximation of the proximal operator in a zeroth-order manner (Osher et al., 2023; Tibshirani et al., 2025; Meng et al., 2025; Zhang et al., 2025).

This approach is particularly attractive because it *only requires function evaluations* and avoids the need for derivatives or closed-form solutions. Subsequent research has investigated HJ-Prox applications, primarily in global optimization via adaptive proximal point algorithms (Heaton et al., 2024; Zhang et al., 2024). These applications, however, remain limited to the proximal point algorithm, which is gradient descent on the Moreau envelope of $f$, leaving the broader family of splitting algorithms unexplored. Our work expands upon the theory of HJ-Prox by creating a comprehensive framework that can be applied to the entire family of splitting algorithms for convex optimization, including the proximal point method (PPM), proximal gradient descent (PGD) (Rockafellar, 1970; Ryu & Yin, 2022), Douglas Rachford Splitting (DRS) (Lions & Mercier, 1979; Eckstein & Bertsekas, 1992), Davis-Yin Splitting (DYS) (Davis & Yin, 2017), and primal-dual hybrid gradient (PDHG) (Chambolle & Pock, 2011). To our knowledge, the direct approximation of the proximal operator via HJ equations for use in

general splitting methods has not been previously explored.

## 3. HJ-Prox-based Operator Splitting

We now show how HJ-Prox can be incorporated into splitting algorithms such as PGD, DRS, DYS, and PDHG. The key idea is simple: by replacing exact proximal steps with their HJ-Prox approximations, we retain convergence guarantees while eliminating the need for closed-form proximal formulas or costly inner optimization loops. For readability, all proofs are provided in the Appendix.

### 3.1. Existing Results

Our analysis builds on a classical result concerning perturbed fixed-point iterations. Theorem 5.2 in (Combettes, 2001) established convergence of Krasnosel'skiĭ–Mann (KM) iterations subject to summable errors:

**Theorem 3.1** (Convergence of Perturbed Krasnosel'skiĭ-Mann Iterates (Combettes, 2001))**.** *Let* $\{x_k\}_{k \geq 0}$ *be a sequence in* $\mathbb{R}^n$ *generated by the iteration*

$$x_{k+1} = T_k x_k + \epsilon_k, \quad (5)$$

*where each* $T_k \colon \mathbb{R}^n \to \mathbb{R}^n$ *belongs to the class of averaged operators, and the solution set* $S = \bigcap_{k \geq 0} \text{Fix} \, T_k$ *is nonempty. If the error sequence is summable (* $\sum_{k=0}^{\infty} \|\epsilon_k\| < \infty$ *) and the operators satisfy a closedness condition such that every cluster point of* $\{x_k\}$ *lies in* $S$*, then* $\{x_k\}_{k \geq 0}$ *converges to a point* $x^* \in S$*.*

Thus, to establish convergence of HJ-Prox–based splitting, it suffices to bound the HJ approximation error. The following result provides the required bound.

**Theorem 3.2** (Error Bound on HJ-Prox (Crandall & Lions, 1983))**.** *Let* $f : \mathbb{R}^n \mapsto \mathbb{R}$ *be LSC and convex. Then the Hamilton-Jacobi approximation incurs errors that are uniformly bounded.*

$$\sup_x \left\| \text{prox}_{tf}^{\delta}(x) - \text{prox}_{tf}(x) \right\| \leq \sqrt{nt\delta}. \quad (6)$$

This result was originally proved in (Crandall & Lions, 1983) in the context of viscosity solutions of Hamilton-Jacobi PDEs (and later in (Zhang et al., 2024; Osher et al., 2023; Darbon & Langlois, 2021; Darbon et al., 2023) in the context of proximals). This uniform error bound guides the choice of $\delta$ in each iteration of our splitting algorithms. In particular, by selecting $\delta_k$ so that the resulting error sequence is summable, Theorem 3.1 ensures convergence of the HJ-Prox–based methods if integrals are evaluated exactly (Di et al., 2025).

### 3.2. Convergence Analysis of HJ-Prox-based Splitting

Theorem 3.2 *assumes exact integral evaluation*, while in practice we use Monte Carlo sampling to approximate inte-

grals. This introduces additional error which we bound in the following result.

**Theorem 3.3** (Monte Carlo Bound on HJ-Prox). *Let $f : \mathbb{R}^n \mapsto \mathbb{R}$ be convex, LSC and $L$-Lipschitz. Let $\widehat{\text{prox}}_{tf}^{\delta}(x)$ denote the Hamilton-Jacobi approximation with finite sample size $N$ and $e(x) = \widehat{\text{prox}}_{tf}^{\delta}(x) - \text{prox}_{tf}(x)$ denote its approximation error. For $\alpha \in (0,1)$ and $N \geq \frac{8J_\star}{\alpha}$, we have the following probabilistic error bound.*

$$P\left( \|e(x)\| > \sqrt{\frac{8J_\star M_\star}{\alpha N}} + \sqrt{nt\delta} \right) \leq \alpha, \qquad (7)$$

*where $J_\star = \exp\left(\frac{2}{\delta}L^2 t\right)$ and $M_\star = nt\delta + (2\sqrt{nt\delta} + 3Lt)^2$.*

Theorem 3.1 is deterministic and requires a summable error sequence. Since HJ–Prox introduces random errors, we verify that summability holds almost surely and then invoke Theorem 3.1 pathwise. To this end, we must require restrictions (at iteration $k$) on the smoothness parameter $\delta_k$, the number of samples $N_k$, and the proximal parameter $t_k$, and the tail probability bound $\alpha_k$.

**Assumption 3.4.** Assume the following.

1. The sequence $t_k > 0$ converges to 0 at rate $\mathcal{O}(1/k)$.
2. The sequences $\delta_k > 0$ and $\alpha_k \in (0,1)$ satisfy $\sum_{k=1}^{\infty} \delta_k < \infty$ and $\sum_{k=1}^{\infty} \alpha_k < \infty$.
3. The sequence $N_k$ satisfies both $N_k \geq \frac{8J_k}{\alpha_k}$ and $\sum_{k=1}^{\infty} \sqrt{\frac{8J_k M_k}{\alpha_k N_k}} + \sqrt{n\, t_k\, \delta_k} < \infty$ where $J_k$ and $M_k$ are the constants in Theorem 3.3 evaluated at $(t_k, \delta_k)$.

We rely on Theorem 3.1 to prove the convergence of the four HJ-Prox-based operator splitting methods. Associated with each algorithm of interest is an algorithm map $T_k$ that takes the current iterate $x_k$ to the next iterate $x_{k+1}$. The key idea is to show that $T_k$ for PPM and PGD satisfy the conditions in Theorem 3.1. And in particular, the almost sure summability of the errors when using HJ-Prox can be guaranteed via Theorem 3.3.

**Theorem 3.5** (HJ-Prox PPM). *Let $f, g$ be proper, LSC, convex and $L$-Lipschitz. Consider the HJ-Prox-based PPM iteration given by*

$$x_{k+1} = \widehat{\text{prox}}_{t_k(f+g)}^{\delta_k}(x_k), \qquad (8)$$

*with parameters satisfying the conditions of Assumption 3.4. Then $x_k$ converges almost surely to a minimizer of $f + g$.*

**Theorem 3.6** (HJ-Prox PGD). *Let $f, g$ be proper, LSC, convex and $L$-Lipschitz, with $f$ additionally $L'$-smooth. Consider the HJ-Prox-based PGD iteration given by*

$$x_{k+1} = \widehat{\text{prox}}_{t_k g}^{\delta_k}(x_k - t_k \nabla f(x_k)), \qquad (9)$$

*with step size $0 < t_k < 1/L'$ and parameters satisfying the conditions of Assumption 3.4. Then $x_k$ converges almost surely to a minimizer of $f + g$.*

An example of parameter choices satisfying Assumption 3.4 is given by $\delta_k = \frac{1}{k^{p+1}}, \quad \alpha_k = \frac{1}{k^{p+2}}, \quad t_k = \frac{1}{k}$, with $p > 0$. Allowing $t_k \to 0$ plays a key role in controlling the required sample complexity. In particular, the number of samples required at iteration $k$ satisfies $N_k = \mathcal{O}\left(e^{k^p} k^{p+2}\right)$, which reduces the growth of the sample complexity from exponential to subexponential in $k$. Although the exponential factor ultimately dominates asymptotically, for values of $p$ close to zero the term $e^{k^p}$ grows extremely slowly. As a result, over a wide range of practically relevant iterations, the overall sample complexity exhibits behavior that is effectively polynomial.

As DRS, DYS, and PDHG are more complex splitting methods, $t$ must remain constant at each iteration in order to satisfy the requirement that $S$ be non-empty in Theorem 3.1. We therefore use a slightly different assumption for these algorithms.

**Assumption 3.7.** Fix $t > 0$ and assume the following.

1. The sequences $\sqrt{\delta_k} > 0$ and $\alpha_k \in (0,1)$ satisfy $\sum_{k=1}^{\infty} \sqrt{\delta_k} < \infty$ and $\sum_{k=1}^{\infty} \alpha_k < \infty$.
2. The sequence $N_k$ satisfies both $N_k \geq \frac{8J_k}{\alpha_k}$ and $\sum_{k=1}^{\infty} \sqrt{\frac{8J_k M_k}{\alpha_k N_k}} + \sqrt{n\, t\, \delta_k} < \infty$ where $J_k$ and $M_k$ are the constants in Theorem 3.3 evaluated at $\delta_k$.

As in the proofs of PGD and PPM, these assumptions ensure conditions needed to apply Theorem 3.1.

**Theorem 3.8** (HJ-Prox DRS). *Let $f, g$ be proper, convex, LSC, and $L$-Lipschitz. Consider the HJ-Prox–based DRS iteration given by*

$$
\begin{aligned}
x_{k+1/2} &= \widehat{\text{prox}}_{tf}^{\delta_k}(z_k), \\
x_{k+1} &= \widehat{\text{prox}}_{tg}^{\delta_k}(2x_{k+1/2} - z_k), \qquad (10) \\
z_{k+1} &= z_k + x_{k+1} - x_{k+1/2},
\end{aligned}
$$

*with parameters satisfying the conditions of Assumption 3.7. Then $x_k$ converges almost surely to a minimizer of $f + g$.*

**Theorem 3.9** (HJ-Prox DYS). *For DYS, consider $f + g + h$. Let $f, g, h$ be proper, LSC, convex and $L$-Lipschitz, with $h$ additionally $L'$-smooth. Consider the HJ-Prox–based DYS algorithm given by*

$$
\begin{aligned}
y_{k+1} &= \widehat{\text{prox}}_{tf}^{\delta_k}(x_k), \\
z_{k+1} &= \widehat{\text{prox}}_{tg}^{\delta_k}\left(2y_{k+1} - x_k - t\nabla h(y_{k+1})\right) \qquad (11) \\
x_{k+1} &= x_k + z_{k+1} - y_{k+1}
\end{aligned}
$$

*with parameters satisfying the conditions of Assumption 3.7 and $0 < t < 2/L'$. Then $x_k$ converges almost surely to a minimizer of $f + g + h$.*

**Theorem 3.10** (HJ-Prox PDHG). *Let $f, g$ be proper, convex, and LSC. Consider the HJ-Prox–based PDHG algorithm*

*given by*

$$
\begin{aligned}
y_{k+1} &= \widehat{\text{prox}}_{\sigma g^*}^{\delta_k}(y_k + \sigma A x_k), \\
x_{k+1} &= \widehat{\text{prox}}_{\tau f}^{\delta_k}(x_k - \tau A^\top y_{k+1}),
\end{aligned}
\tag{12}
$$

*with parameters $\tau, \sigma > 0$ satisfying $\tau \sigma \|A\|^2 < 1$ and the conditions of $t$ in Assumption 3.7, where $g^*$ denotes the Fenchel conjugate of $g$. Then $x^k$ converges almost surely to a minimizer of $f(x) + g(Ax)$.*

### 3.3. Why Splitting?

Since HJ-PPM can be applied to general objectives, one might be tempted to ignore alternative splittings such as HJ-PGD, HJ-DRS, HJ-DYS, and HJ-PDHG altogether. However, theoretical results show that the Monte Carlo error associated with the HJ-Prox approximation depends exponentially on the squared Lipschitz constant of the function being approximated. The constant in Theorem 3.3,

$$
J_\star = \exp\left(\frac{2L^2 t}{\delta}\right),
\tag{13}
$$

governs the sample complexity required to control the approximation error. Consequently, reducing the effective Lipschitz constant entering the HJ-Prox approximation has a dramatic impact on both the variance of the estimator and the resulting theoretical computational cost. When HJ-Prox is applied directly to a composite objective of the form $f(x) + g(x)$, the Lipschitz constant satisfies

$$
L_{f+g} \leq L_f + L_g,
\tag{14}
$$

whenever $f$ and $g$ are $L_f$- and $L_g$-Lipschitz, respectively. The corresponding worst-case Monte Carlo constant is therefore bounded by

$$
J_{f+g} = \exp\left(\frac{2L_{f+g}^2 t}{\delta}\right)
\tag{15}
$$

$$
\leq \exp\left(\frac{2(L_f + L_g)^2 t}{\delta}\right)
$$

$$
= \exp\left(\frac{2(L_f^2 + L_g^2 + 2L_f L_g)\, t}{\delta}\right).
\tag{16}
$$

By contrast, operator splitting decouples the proximal evaluations, enabling HJ-Prox to be applied separately to $f$ and $g$ (or only to one of the functions if the other has an explicit proximal formula). This results in independent Monte Carlo constants

$$
J_f = \exp\left(\frac{2L_f^2 t}{\delta}\right), \qquad J_g = \exp\left(\frac{2L_g^2 t}{\delta}\right),
\tag{17}
$$

with a total approximation complexity proportional to $J_f + J_g$. Operator splitting substantially reduces the error resulting from $J_{f+g}$ and yields a tighter theoretical bound (especially if one of the proximals has a closed form solution)

on the Monte Carlo error. The benefits of applying HJ-Prox in a selective manner within operator splitting algorithms are demonstrated empirically in Figure 4 and explained in Section 4.3.

### 3.4. HJ-Prox Discussion

The above theory illustrates how HJ-Prox can be seamlessly integrated into a broad class of operator splitting algorithms by treating each method as a perturbed fixed-point iteration and controlling the approximation error via classical Krasnosel'skiĭ–Mann theory. Crucially, exact proximal operators are not required for convergence; if the HJ-Prox approximation errors are summable almost surely, standard splitting methods retain their global convergence guarantees.

The smoothness parameter $\delta_k$ governs a fundamental trade-off between bias and stability: decreasing $\delta_k$ reduces smoothing bias and improves asymptotic accuracy, but overly small values can lead to numerical instability. From a theoretical perspective, HJ-PPM and HJ-PGD require only subexponential sample complexity, whereas HJ-DRS, HJ-DYS, and HJ-PDHG require exponential sample complexity. However, we find in practice that fixing the number of samples per iteration and choosing a moderately small $\delta_k$ with a slowly decreasing schedule yields stable convergence across our benchmarks, despite the conservatism of the sufficient conditions in Assumptions 3.4 and 3.7.

## 4. Experiments

We evaluate HJ-Prox as a drop-in replacement for proximal operators within standard splitting methods PGD, DRS, DYS, and PDHG. We consider seven convex nonsmooth optimization problems that span settings with closed-form proximal operators, problems that require specialized numerical routines, and hybrid configurations. For fair comparison, all HJ-Prox and exact-prox baselines use identical objective functions, step sizes, and algorithmic parameters where applicable. HJ-Prox methods use a decreasing $\delta_k$ schedule (guided by Assumption 3.7) and a fixed Monte Carlo sample size $N = 1000$ and $t$. While our convergence theory assumes growing sample sizes $N_k$ to ensure almost sure summability of approximation errors, we empirically observe robust convergence performance with fixed $N$. We compare recovered solutions against analytical based methods and report objective values versus iteration. The final objective is shown in each legend.

Across all experiments, HJ-Prox recovers solutions visually indistinguishable from analytical baselines, validating that zeroth-order approximations do not compromise solution quality in standard splitting algorithms. Further experimental details are displayed in H.

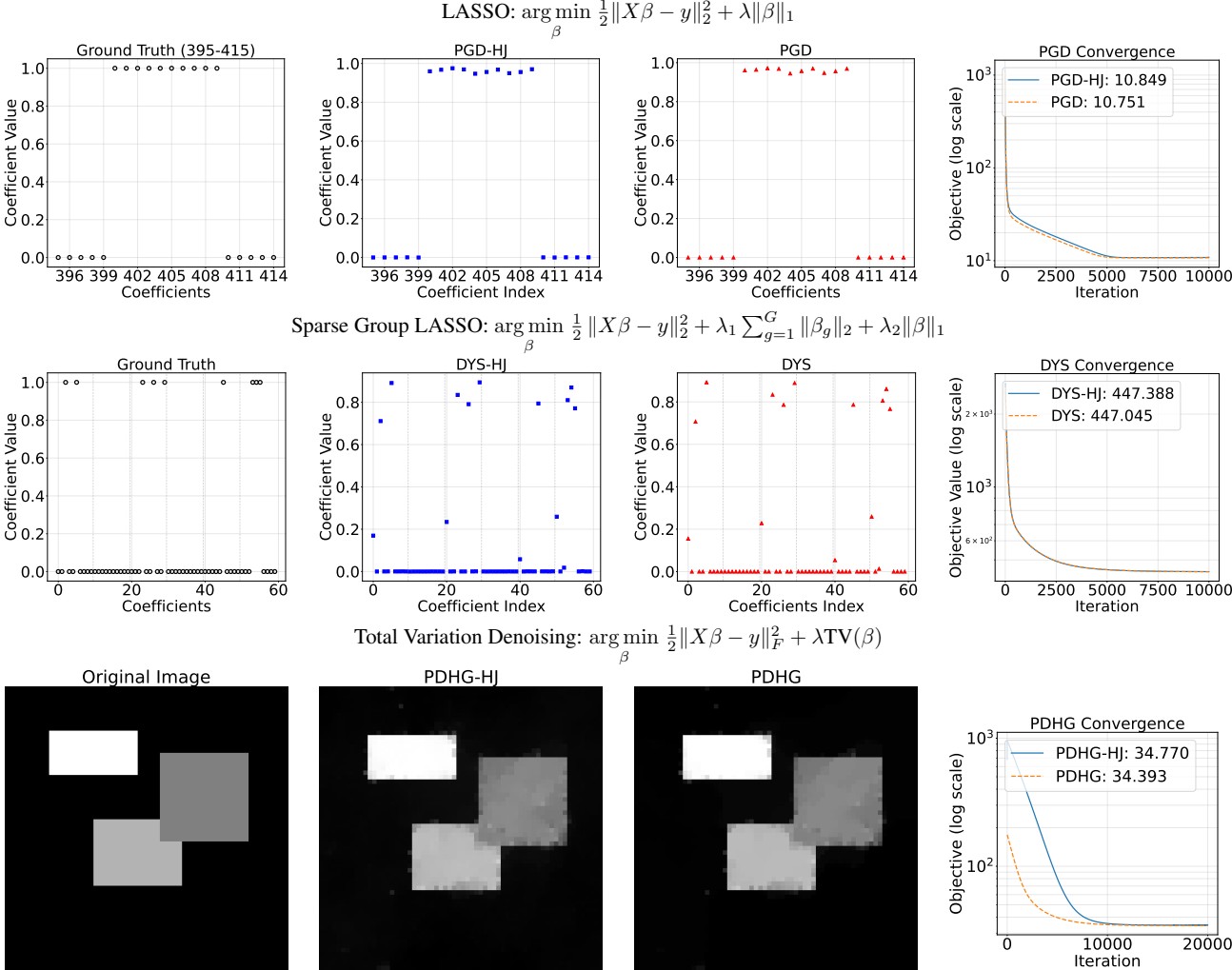

*Figure 1.* Top row: LASSO experiment solved using PGD. HJ-Prox is called on the $\ell_1$ regularizer. Middle row: Sparse Group LASSO using DYS. HJ-Prox is called on the Group and $\ell_1$ regularizer. Bottom row: Total Variation Denoising solved with PDHG. HJ-Prox is called on the Total Variation penalty. All examples feature closed-form projection-based proximal operators. While final reconstructions are visually identical, the convergence rates differ, with the more difficult TV objective requiring increased iterations for the HJ-Prox variant.

### 4.1. Comparison on Problems with Closed-Form Proximal Steps

We first validate HJ-Prox on problems with closed-form or simple projection-based proximal operators. We apply PGD to the LASSO problem, where the baseline proximal operator is standard soft-thresholding, and Davis–Yin splitting to non-overlapping Sparse Group LASSO, where exact updates involve closed-form groupwise and elementwise shrinkage. We further consider Total Variation (TV) denoising using PDHG, for which the exact proximal updates reduce to simple projections onto box constraints.

For LASSO and sparse group LASSO, we perform effective variable selection with both approaches, shrinking true zero coefficients toward zero. We observe that computational cost varies by problem structure. Since the HJ-Prox error depends on the parameter $t$ as derived in Theorem 3.3, smaller step sizes are often required to minimize error and maintain high precision. Consequently, this calls for a higher number of iterations to reach convergence. This trade-off is especially obvious in the Total Variation denoising problem, which typically requires significantly more iterations to match the analytical baseline compared to simpler tasks like LASSO. HJ-Prox precisely recovers the solutions obtained via exact closed-form proximal operators, demonstrating that the zeroth-order approximation achieves near same accuracy as analytical methods. Results for these experiments are shown in Figure 1.

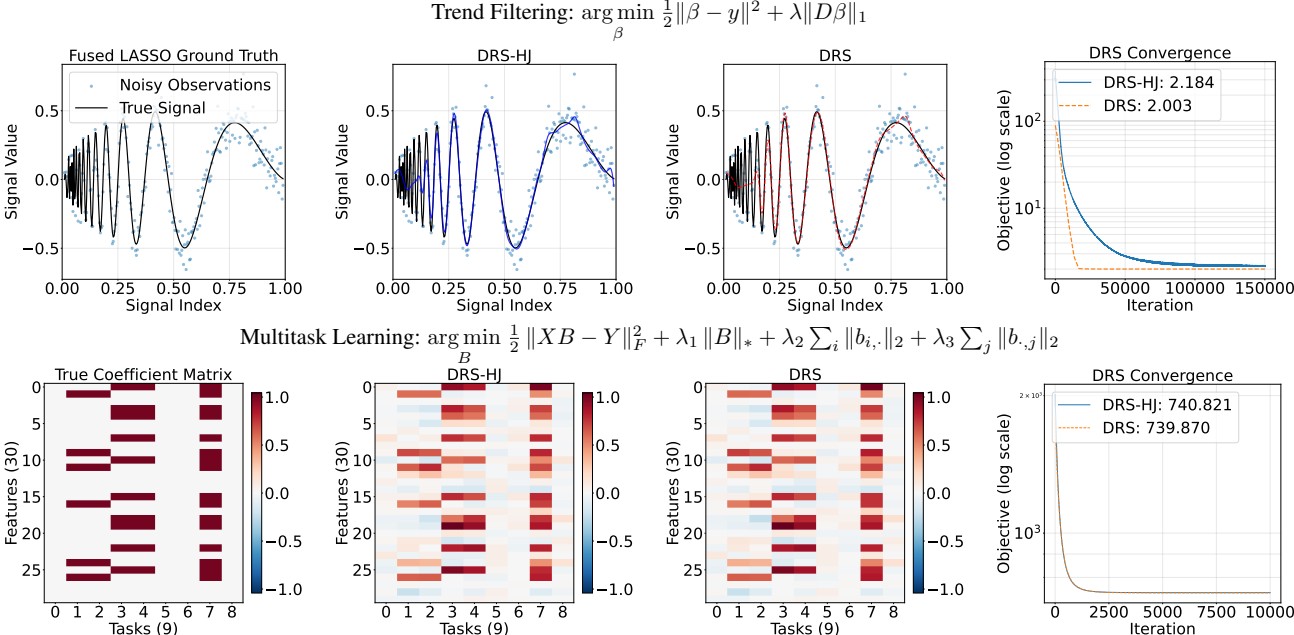

*Figure 2.* Top row: Trend Filtering experiment solved using DRS. HJ-Prox is called on the differencing matrix. Bottom row: Multitask Learning using DRS. HJ-Prox is called on the clumped data fidelity term and onto the row and column penalties. Results confirm that HJ-Prox matches the convergence of complex baseline solvers, both of which require inner loops or problem reformulations.

## 4.2. Comparison on Problems Without Closed-Form Proximals

Next, we consider problems where proximal operators are well defined but require specialized or computationally intensive routines to evaluate. For multitask learning, we apply DRS to a structured matrix regularizer. The analytical baseline requires iterative *full singular value decompositions (SVD)* to reconstruct the matrix, as well as Dykstra's algorithm to handle mixed-norm penalties. For trend filtering, the exact method uses a product-space reformulation that requires solving dense linear systems via Cholesky factorization at every iteration.

In both cases, HJ-Prox serves as a function-based drop-in replacement that avoids specialized inner solvers, significantly simplifying implementation. In multitask learning, the HJ-Prox iterates closely match the analytical updates. Crucially, while HJ-Prox requires singular values for evaluation, it circumvents computing a full SVD. For trend filtering, HJ-Prox requires a larger number of iterations to converge, consistent with the increased difficulty of the proximal operator associated with the regularization term. Importantly, HJ-Prox enables us to apply splitting algorithms directly to the original objective function, completely eliminating the need for current complex reformulations, such as lifting the problem into a product space or designing specialized iterative subroutines, thereby highlighting a streamlined universal approach to nonsmooth optimization. Results for these experiments are shown in Figure 2.

## 4.3. Splitting vs Non-splitting

To demonstrate the necessity of operator splitting, we first compare splitting schemes (PGD, DRS) against the non-splitting Proximal Point Method (PPM). PGD and DRS, decompose the problem by applying individual proximal or gradient steps to $f$ and $g$ separately within each iteration. In contrast, the non-splitting HJ-PPM attempts to compute the HJ-Prox of the combined function $f + g$ directly. All methods use identical parameters for a fair comparison: $t$, $\delta$, and $N$ as seen in Figure 3. A notable methodological difference is that PPM does not admit a clear, analogous step-size parameter for balancing the two objectives, as it treats them as a single entity. To ensure a competitive baseline, we manually tuned the PPM step size for each experiment.

As shown in Figure 3, HJ-PPM consistently underperforms splitting-based methods. It exhibits slower convergence rates and converges to higher final objective values across experiments. This performance gap stems from a key disadvantage highlighted in Section 3.3. The results establish that a composite, non-splitting approach is computationally less efficient and fails to match the benefits of operator splitting.

Moreover, a key practical advantage unlocked by operator splitting is the ability to deploy *hybrid strategies* in our framework. This flexibility is crucial for problems where the composite objective comprises terms of mixed tractability. For instance, one term (like an $\ell_1$-regularizer) often admits a

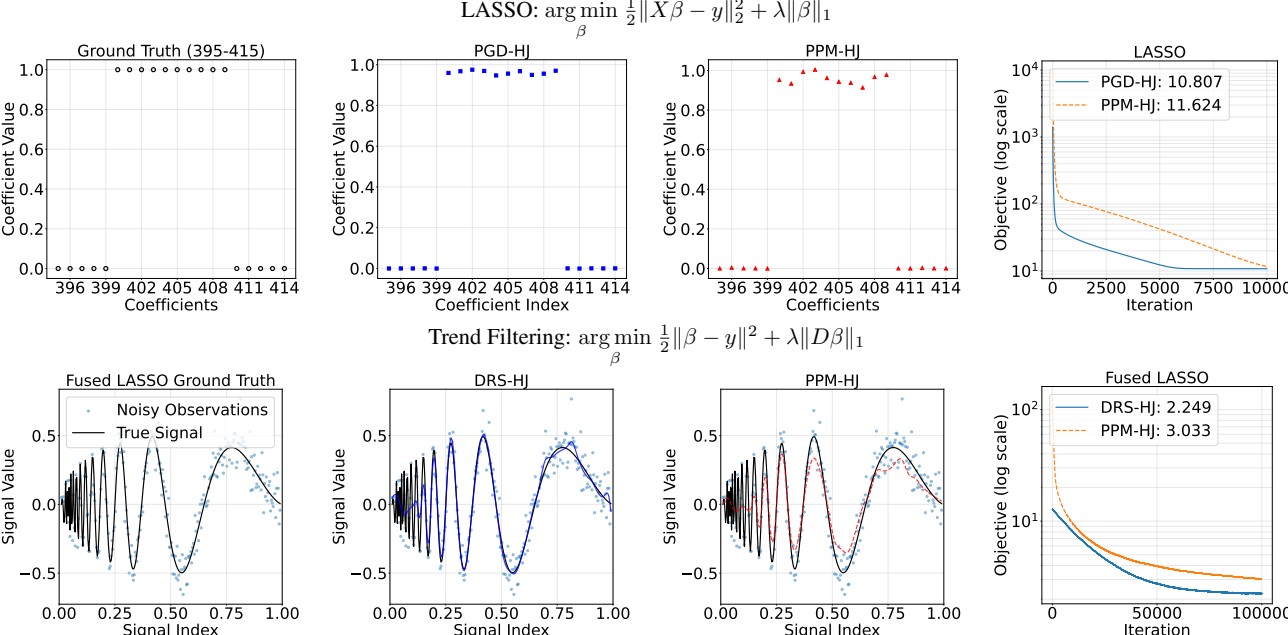

*Figure 3.* Top row: HJ-PPM struggles to match convergence of HJ-PGD. Bottom row: HJ-PPM struggles to converge to similar solutions as opposed to using splitting methods. These experiments highlight that HJ-PPM is computationally less efficient and lacks the robust error control of the alternative methods.

cheap, closed-form proximal operator. Splitting frameworks enable us to apply *this exact proximal operator for the proximable term*, while restricting the HJ-Prox approximation exclusively to the truly non-proximable (i.e., terms without closed-form proximals) component and eliminating approximation error for the analytical steps. Consequently, the overall algorithmic error is no longer an accumulation of errors from multiple approximate operations but is determined by the error introduced in approximating the non-proximal portion with HJ-Prox. This leads to more accurate convergence compared to a fully approximate splitting method. We concretely illustrate this performance hierarchy using a non-negative LASSO problem in Figure 4. We compare four configurations under identical algorithmic parameters. First, by bringing the constraint to the objective function as an indicator function, the fully analytical DYS method serves as the gold standard, which employs exact proximal operators for both $\ell_1$ and indicator function. Second, a hybrid variant (DYS-HJ-1) applies the HJ-Prox approximation *only to $\ell_1$*. Third, a fully approximate method (DYS-HJ-2) *replaces both proximal operators* with HJ-Prox approximations. Finally, the non-splitting HJ-PPM acts as the composite baseline, applying HJ-Prox directly to the full objective.

To provide a fair performance comparison, we use the fixed-point residual of the fully analytic DYS algorithm, which serves as the correct KKT conditions for this problem. The results reveal that performance degrades progressively as more exact proximal operators are replaced with approxi-

mations. The hybrid method (DYS-HJ-1) significantly outperforms the fully approximate one (DYS-HJ-2), which demonstrates the value of preserving exact computations where possible. Crucially, the non-splitting HJ-PPM performs worst as it achieves the highest objective value and more importantly, the largest fixed-point residual.

### 4.4. Overlapping Group LASSO Gene Expression

We evaluate our approach on the GSE2034 breast cancer gene expression dataset (Wang et al., 2005), a standard benchmark for high-dimensional, low-sample-size prediction in genomics. We organize the features using KEGG pathways, yielding 298 overlapping groups that cover 41% of the 13237 measured genes.

We impose both an $\ell_1$ and overlapping group LASSO penalty to promote pathway-level sparsity while allowing genes to participate in multiple pathways. From an optimization perspective, unlike the standard group LASSO the resulting regularizer is not block separable, which renders the proximal operator *unavailable in closed form*. Existing approaches, such as FoGLASSO (Yuan et al., 2011), address this difficulty through dual reformulations and specialized solvers. In contrast, HJ-Prox provides a direct approximation of the challenging proximal operator, enabling the use of standard operator splitting methods in the primal problem without variable duplication or dual reformulation, while attaining results comparable to FoGLASSO in Figure 5.

Non-Negative LASSO: $\arg\min_{\beta} \frac{1}{2}\|X\beta - y\|_2^2 + \lambda\|\beta\|_1$ s.t $\beta \geq 0$

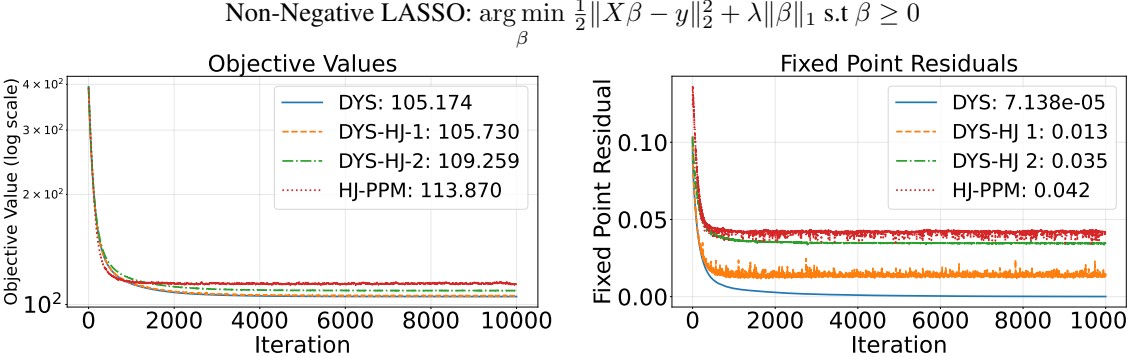

*Figure 4.* Under a fixed $\delta$, pure PPM decreases the objective rapidly but is limited to a solution neighborhood with high error floors. The results further demonstrate that a single call of HJ-Prox outperforms two calls, suggesting a hybrid framework can be optimal when applicable. All fixed-point residuals are calculated using the analytical fixed-point operator to ensure fair comparison across methods.

Overlapping Group LASSO: $\arg\min_{\beta} \frac{1}{2}\|X\beta - y\|_2^2 + \lambda_1 \sum_{g=1}^{G} w_g\|\beta_g\|_2 + \lambda_2\|\beta\|_1$

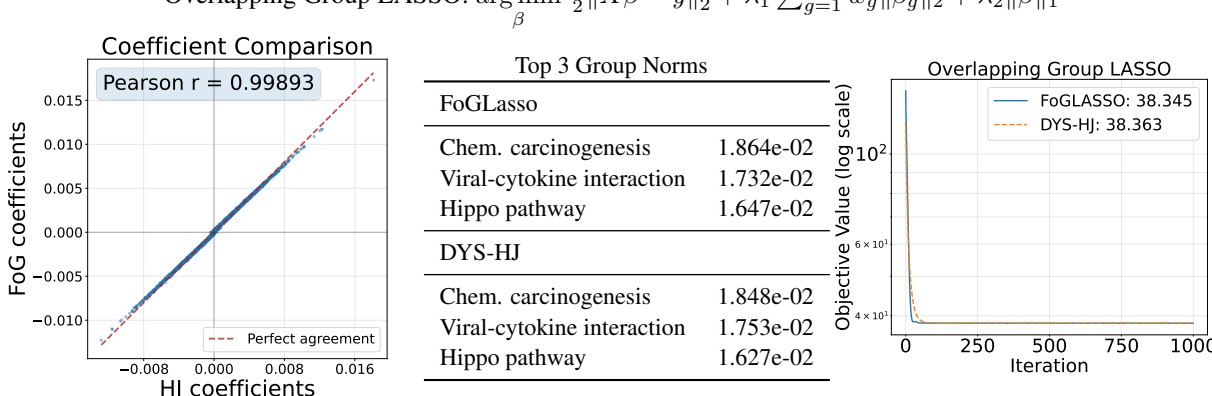

*Figure 5.* Although similar in structure to the Group LASSO experiment in Figure 1, we allow the $\beta$ groupings to overlap here. We show convergence in log scale of the objective functions. We utilize an adaptive DYS here (Pedregosa & Gidel, 2018). DYS-HJ is run with a decreasing $\delta$-schedule and fixed $N = 1000$. FoGLasso is implemented exactly as in (Yuan et al., 2011). We run for 100000 iterations but zoom in to show the relevant iteration window for comparable convergence progress. The coefficient vector of length 13237 has high correlation between FoGLASSO and DYS-HJ, and the group norms are numerically close together.

## 5. Limitations and Future Work

A gap persists between the conservative sample complexity required by our theory and the practical efficiency observed in experiments. While our analysis rigorously accounts for Monte Carlo error to provide convergence guarantees, the empirical success of $\delta$ and $N$ suggests an opportunity for tighter theoretical bounds.

Currently, our framework uses predetermined schedules for the smoothing parameter $\delta$ and maintains a fixed sample size $N$ throughout optimization. While this approach is simple to implement and performs well empirically, adaptive strategies based on iteration-specific information could potentially improve convergence rates, particularly for challenging problem instances. Future work will focus on adaptive splitting algorithms that jointly optimize $N$ and $\delta$ based on iteration dynamics, potentially integrating these methods within a Learning-to-Optimize approach (Chen et al., 2022;

Heaton & Fung, 2023; Mckenzie et al., 2024; McKenzie et al., 2024) to enable robust, automatic parameter tuning.

## 6. Conclusion

Our work demonstrates that HJ-Prox can be successfully integrated into operator splitting frameworks while maintaining theoretical convergence guarantees, providing a generalizable method for solving composite convex optimization problems. By replacing exact proximal operators with a zeroth-order Monte Carlo approximation, we have established that algorithms such as PGD, DRS, DYS, and the PDHG method retain their convergence properties under mild conditions. This framework offers practitioners a universal approach to solving difficult nonsmooth optimization, reducing their reliance on complex proximal computations. Code for this work can be found at `https://github.com/nicholasdi2000/HJ-Splitting`.

## Acknowledgments

The authors thank Howard Heaton for fruitful discussions.

## Impact Statement

This paper contributes advances to optimization methods in machine learning. By providing a general framework for approximating proximal operators, the work lowers the barrier for data scientists and statisticians to optimize complex objective functions without requiring closed-form proximal calculus or specialized mathematical derivations. We do not anticipate direct negative societal impacts arising from this work.

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

# A. Proof of HJ-Prox Error Bound

For completeness and ease of presentation, we restate the Theorem 3.2.

*Let $f : \mathbb{R}^n \mapsto \mathbb{R}$ be LSC and convex. Then the Hamilton-Jacobi approximation incurs errors that are uniformly bounded.*

$$\sup_x \left\| \mathrm{prox}_{tf}^\delta(x) - \mathrm{prox}_{tf}(x) \right\| \leq \sqrt{n t \delta}. \tag{18}$$

*Proof.* Fix the parameters $t$ and $\delta$. For notational convenience, denote $\mathrm{prox}_{tf}(x)$ by $z^\star(x)$ and $\mathrm{prox}_{tf}^\delta(x)$ by $z_\delta(x)$. Let

$$\phi_x(z) = f(z) + \frac{1}{2t} \|x - z\|^2. \tag{19}$$

Making the change of variable $w = z - z^\star(x)$ in (4) enables us to express the approximation error as

$$z_\delta(x) - z^\star(x) = \frac{\int w \exp\left(-\frac{\phi_x(z^\star(x)+w)-\phi_x(z^\star(x))}{\delta}\right) dw}{\int \exp\left(-\frac{\phi_x(z^\star(x)+w)-\phi_x(z^\star(x))}{\delta}\right) dw}. \tag{20}$$

Let

$$Z_\delta = \int \exp\left(-\frac{\phi_x(z^\star(x) + w) - \phi_x(z^\star(x))}{\delta}\right) dw \tag{21}$$

and

$$g(w) = \phi_x(z^\star(x) + w) - \phi_x(z^\star(x)). \tag{22}$$

Then

$$\rho_\delta(w) = \frac{e^{-\frac{g(w)}{\delta}}}{Z_\delta} \tag{23}$$

defines a proper density.

Equations (20) and (23) together imply that the approximation error can be written as the expected value of a continuous random variable $W$ whose probability law has the density $\rho_\delta$,

$$z_\delta(x) - z^\star(x) = \int w \, \rho_\delta(w) dw = \mathbb{E}_{\rho_\delta}(W). \tag{24}$$

Taking the norm of both sides of (24) leads to a bound on the norm of the approximation error.

$$\|z_\delta(x) - z^\star(x)\| = \|\mathbb{E}_{\rho_\delta}(W)\| \leq \mathbb{E}_{\rho_\delta}(\|W\|) \leq \sqrt{\mathbb{E}_{\rho_\delta}(\|W\|^2)}. \tag{25}$$

The first inequality is due to Jensen's inequality since norms are convex. The second is due to the Cauchy-Schwarz inequality. Our goal is to show that

$$\sqrt{\mathbb{E}_{\rho_\delta}(\|W\|^2)} \leq \sqrt{n t \delta}, \tag{26}$$

since inequalities (25) and (26) together imply that

$$\|z_\delta(x) - z^\star(x)\| \leq \sqrt{n t \delta}. \tag{27}$$

We prove (26) in two steps. We first show that

$$\mathbb{E}_{\rho_\delta}(\|W\|^2) \leq t \mathbb{E}_{\rho_\delta}(\langle W, \nabla g(W)\rangle), \tag{28}$$

where $g$ is the convex function defined in (22). We then show that

$$\mathbb{E}_{\rho_\delta}\left(\langle W, \nabla g(W)\rangle\right) = n\delta. \tag{29}$$

Before proceeding to prove these steps, we address our abuse of notation in (28) and (29). Although $\nabla g$ may not exist everywhere, it exists almost everywhere. Recall that $g$ is locally Lipschitz because it is convex. Furthermore, any locally Lipschitz function is differentiable almost everywhere by Rademacher's theorem. Hence $\nabla g$ exists almost everywhere. Consequently, the expectation $\mathbb{E}_{\rho_\delta}\langle W, \nabla g(W)\rangle$ is well defined.

To show (28), first note that by Fermat's rule, $0 \in \partial \phi_x(z^\star(x))$ where $\partial \phi(z)$ denotes the subdifferential of $\phi$ at $z$. Consequently, for any $z \in \mathbb{R}^n$

$$
\begin{aligned}
\phi_x(z) &\geq \phi_x(z^\star(x)) + \langle 0, z - z^\star(x)\rangle + \frac{1}{2t}\|z - z^\star(x)\|^2 \\
&= \phi_x(z^\star(x)) + \frac{1}{2t}\|z - z^\star(x)\|^2,
\end{aligned}
\tag{30}
$$

since $\phi_x(z)$ is $\frac{1}{t}$-strongly convex. Plugging $z = z^\star(x) + w$ into inequality (30) implies that

$$g(w) \geq \frac{1}{2t}\|w\|^2. \tag{31}$$

We also know that since $\phi_x(z)$ is $\frac{1}{t}$-strongly convex, so is $g(z)$, since $g$ is a translation of $\phi$ plus a constant shift. For $\frac{1}{t}$ strongly convex $g$ with $\nabla g(w)$ existing at point $w$, and setting $w' = 0$, the strong convexity inequality gives

$$g(0) \geq g(w) + \langle \nabla g(w), 0 - w\rangle + \frac{1}{2t}\|0 - w\|^2. \tag{32}$$

Therefore,

$$\langle \nabla g(w), w\rangle \geq g(w) + \frac{1}{2t}\|w\|^2. \tag{33}$$

Using (31) and (33),

$$\langle \nabla g(w), w\rangle \geq \frac{1}{t}\|w\|^2, \tag{34}$$

which implies that

$$\frac{1}{t}\int \|w\|^2 \rho_\delta(w)dw \leq \int \langle w, \nabla g(w)\rangle \rho_\delta(w)dw. \tag{35}$$

To show (29), consider $w$ where $\nabla g$ exists and let $h(w) = \exp(-g(w)/\delta)$. By the chain rule

$$\frac{\partial}{\partial w_j}g(w)h(w) = -\delta\frac{\partial}{\partial w_j}h(w). \tag{36}$$

Integrating both sides of (36) over $\mathbb{R}^d$ gives

$$
\begin{aligned}
\int_{\mathbb{R}^n} w_j \frac{\partial}{\partial w_j}g(w)h(w)dw &= -\delta\int_{\mathbb{R}^n} w_j \frac{\partial}{\partial w_j}h(w)dw \\
&= -\delta\int_{\mathbb{R}^{n-1}}\left[\int_{-\infty}^{\infty} w_j \frac{\partial}{\partial w_j}h(w)dw_j\right]dw_{-j},
\end{aligned}
\tag{37}
$$

where $w_{-j}$ is the subvector of $w$ containing all but its $j$th element.

Applying integration by parts on the right hand side of (37) gives

$$\int_{-\infty}^{\infty} w_j \frac{\partial}{\partial w_j}h(w)dw_j = w_j h(w)\Big|_{-\infty}^{\infty} - \int_{-\infty}^{\infty} h(w)dw_j. \tag{38}$$

Note that (31) implies that

$$\lim_{w_j \to \infty} |w_j h(w)| \quad \leq \quad \lim_{w_j \to \infty} \left| w_j e^{-\frac{\|w\|^2}{2t}} \right| \quad = \quad 0. \tag{39}$$

Consequently,

$$\int_{\mathbb{R}^n} w_j \frac{\partial}{\partial w_j} h(w) dw \quad = \quad \int_{\mathbb{R}^{n-1}} \left[ -\int_{-\infty}^{\infty} h(w) dw_j \right] dw_{-j} \quad = \quad -Z_\delta. \tag{40}$$

Equations (37) and (40) together imply that

$$\mathbb{E}_{\rho_\delta} \left( W_j \frac{\partial}{\partial w_j} g(W) \right) \quad = \quad \delta. \tag{41}$$

The linearity of expectations gives (29) completing the proof. $\qquad\square$

## B. Proof of Monte Carlo HJ-Prox Error Bound

For completeness and ease of presentation, we restate the Theorem 3.3.

**Theorem. 3.3.** *Let $f : \mathbb{R}^n \mapsto \mathbb{R}$ be convex, LSC and L-Lipschitz. Let $\widehat{\mathrm{prox}}_{tf}^\delta(x)$ denote the Hamilton-Jacobi approximation with finite sample size $N$ and $e(x) = \widehat{\mathrm{prox}}_{tf}^\delta(x) - \mathrm{prox}_{tf}(x)$ denote its approximation error. For $\alpha \in (0, 1)$ and $N \geq \frac{8 J_\star}{\alpha}$, we have the following probabilistic error bound.*

$$P \left( \|e(x)\| > \sqrt{\frac{8 J_\star M_\star}{\alpha N}} + \sqrt{nt\delta} \right) \quad \leq \quad \alpha, \tag{42}$$

*where $J_\star = \exp \left( \frac{2}{\delta} L^2 t \right)$ and $M_\star = nt\delta + (2\sqrt{nt\delta} + 3Lt)^2$.*

*Proof.* Let

$$\phi_x(z) \quad = \quad f(z) + \frac{1}{2t} \|z - x\|_2^2, \tag{43}$$

and

$$C_\delta^x \quad = \quad \int_{\mathbb{R}^n} \exp \left( -\frac{\phi_x(z)}{\delta} \right) dz. \tag{44}$$

Then

$$\pi(dz) \quad = \quad \frac{\exp \left( -\frac{\phi_x(z)}{\delta} \right)}{C_\delta^x} dz \tag{45}$$

is a probability measure. Let $Z$ be a random variable with law $\pi$. The smoothed proximal point can be interpreted as the expected value under $\pi$. It is hard to directly sample from $\pi$, so we estimate $\mathrm{prox}_{tf}^\delta(x)$ via Self Normalized Importance Sampling under a Gaussian Proposal,

$$q \quad \sim \quad N(x, \sigma^2 I_n), \tag{46}$$
$$\sigma^2 \quad = \quad t\delta. \tag{47}$$

We first decompose our error into a Monte Carlo and deterministic piece via the triangle inequality,

$$\| \widehat{\mathrm{prox}}_{tf}^\delta(x) - \mathrm{prox}_{tf}(x) \| \quad = \quad \| \widehat{\mathrm{prox}}_{tf}^\delta(x) - \mathrm{prox}_{tf}^\delta(x) + \mathrm{prox}_{tf}^\delta(x) - \mathrm{prox}_{tf}(x) \| \tag{48}$$

$$\leq \quad \| \widehat{\mathrm{prox}}_{tf}^\delta(x) - \mathrm{prox}_{tf}^\delta(x) \| + \| \mathrm{prox}_{tf}^\delta(x) - \mathrm{prox}_{tf}(x) \|. \tag{49}$$

We have previously shown that the deterministic part is uniformly bounded by $\sqrt{nt\delta}$. Let $N$ be the sample size used to estimate the proximal operator. For notational purpose, set

$$\mu \;=\; \text{prox}_{tf}^{\delta}(x), \quad \hat{\mu}_N \;=\; \widehat{\text{prox}}_{tf}^{\delta}(x). \tag{50}$$

Define a ratio

$$r(z) \;=\; \frac{\pi(z)}{q(z)}, \tag{51}$$

and quantities which are functions of $Z_i \overset{iid}{\sim} N(x, \sigma^2 I_n)$,

$$U_N \;=\; \frac{1}{N}\sum_{i=1}^{N} r(Z_i)(Z_i - \mu), \quad V_N \;=\; \frac{1}{N}\sum_{i=1}^{N} r(Z_i). \tag{52}$$

Using the ratio, we can write the Monte Carlo error as a fraction

$$\hat{\mu}_N - \mu \;=\; \frac{U_N}{V_N}, \tag{53}$$

Let us define the following events for some $\varepsilon > 0$

$$A \;=\; \left\{ V_N \geq \frac{1}{2} \right\}, \quad B = \{\|\hat{\mu}_N - \mu\|_2 > \varepsilon\}. \tag{54}$$

We control the SNIS bias via a ratio-of-means decomposition, bounding the denominator with Chebyshev and the numerator's second moment by change of measure and Poincaré's inequality. This event split is a common approach used in bounding Monte Carlo Estimates, as seen in Theorem 2.1 (Agapiou et al., 2017), and Section 5.3.2 (Peña et al., 2009). Using the law of total probability,

$$P(B) \;=\; P(A^c \cap B) + P(A \cap B) \;\leq\; P(A^c) + P(A \cap B). \tag{55}$$

Take the expectation of $r(Z)$ with respect to the measure $q$,

$$\mathbb{E}_q[r(Z)] \;=\; \int \frac{\pi(z)}{q(z)} q(z) dz \;=\; 1. \tag{56}$$

Since $r(Z_i)$ are iid,

$$\text{Var}(V_N) \;=\; \frac{1}{N^2}\sum_{i}^{N} \text{Var}\left(r(Z_i)\right) \;=\; \frac{\mathbb{E}_q[r(Z)^2] - 1^2}{N}. \tag{57}$$

Event $A^c$ can be written as a subset

$$A^c \;=\; \{V_N < \frac{1}{2}\} \;\subseteq\; \{|V_N - 1| \geq \frac{1}{2}\}, \tag{58}$$

where we can bound using Chebyshev's inequality

$$P(A^c) \;\leq\; P(|V_N - 1| > \frac{1}{2}) \;\leq\; \frac{\text{Var}(V_N)}{\frac{1}{2}^2} \;<\; \frac{4\mathbb{E}_q[r(Z)^2]}{N}. \tag{59}$$

In event $A \cap B$,

$$\|\hat{\mu}_N - \mu\| \;=\; \left\|\frac{U_N}{V_N}\right\| \;\leq\; 2\|U_N\|. \tag{60}$$

Recall we can rewrite the error as a fraction,

$$B \;=\; \{\|\hat{\mu} - \mu\|^2 > \varepsilon\} \;=\; \left\{\left\|\frac{U_N}{V_N}\right\|^2 > \varepsilon\right\}. \tag{61}$$

When both A and B are true, the intersection of the event is contained in the following subset,

$$A \cap B \quad \subseteq \quad \{4\|U_N\|^2 > \varepsilon\}. \tag{62}$$

which could be bounded using Markov inequality bound,

$$P(A \cap B) \le P(4\|U_N\|^2 > \varepsilon) \quad < \quad \frac{4\mathbb{E}_q[\|U_N\|^2]}{\varepsilon}. \tag{63}$$

Putting together (59) and (63),

$$P(\|\hat{\mu}_N - \mu\|^2 > \varepsilon) \quad \le \quad \frac{4\mathbb{E}_q[r(Z)^2]}{N} + \frac{4\mathbb{E}_q[\|U_N\|^2]}{\varepsilon}. \tag{64}$$

The following 3 lemmas will decompose the second moment of $U_N$ and compute proper upper bounds to necessary components giving us a nice probabilistic statement.

**Lemma B.1** (Second Moment Identity). *The second moment of the numerator $\mathbb{E}_q[\|U_N\|^2]$ can be decomposed into* $\frac{\mathbb{E}_\pi[r(Z)]}{N}\mathbb{E}_{\pi^*}[\|Z - \mu\|^2]$, *where $\pi^*$ is a tilted measure of $\pi$.*

*Proof.* Let us construct a tilted measure

$$\pi^*(z) \quad = \quad \frac{r(z)}{\mathbb{E}_\pi[r(Z)]}\pi(z). \tag{65}$$

Since $Z_i$ for $i \in [N]$ are iid and the expectation of the cross product terms in $U_N$ are 0,

$$\mathbb{E}_q\left[\left\|\frac{1}{N}\sum_i r(Z_i)(Z_i - \mu)\right\|^2\right] \quad = \quad \frac{1}{N^2}\sum_i \mathbb{E}_q\left[\|r(Z_i)(Z_i - \mu)\|^2\right] \tag{66}$$

$$= \quad \frac{\mathbb{E}_q[r(Z)^2\|Z - \mu\|^2]}{N}. \tag{67}$$

Continuing on from equation (67) we can multiply by $\frac{\mathbb{E}_\pi[r(Z)]}{\mathbb{E}_\pi[r(Z)]}$ and algebraically manipulate the expression into two separate expectations,

$$\frac{1}{N}\mathbb{E}_q[r(Z)^2\|Z - \mu\|^2] = \frac{1}{N}\int r(z)^2\|z - \mu\|^2 q(z)dz = \frac{1}{N}\int r(z)\|z - \mu\|^2\pi(z)dz \tag{68}$$

$$= \frac{\mathbb{E}_\pi[r(Z)]}{\mathbb{E}_\pi[r(Z)]}\cdot\frac{1}{N}\int \|z - \mu\|^2 r(z)\pi(z)dz = \mathbb{E}_\pi[r(Z)]\frac{1}{N}\int \|z - \mu\|^2\frac{r(z)}{\mathbb{E}_\pi[r(Z)]}\pi(z)dz \tag{69}$$

$$= \mathbb{E}_\pi[r(Z)]\frac{1}{N}\int \|z - \mu\|^2\pi^*(z)dz = \frac{1}{N}\mathbb{E}_\pi[r(Z)]\mathbb{E}_{\pi^*}[\|Z - \mu\|^2]. \tag{70}$$

$\square$

**Lemma B.2** (Bound on $\mathbb{E}_{\pi^*}[\|Z - \mu\|^2]$). *The expectation of $\|Z - \mu\|^2$ under $\pi^*$ is bounded by $M_\star = nt\delta + (2\sqrt{nt\delta} + 3Lt)^2$*

*Proof.* We decompose $\mathbb{E}_{\pi^*}[\|Z - \mu\|^2]$ using the bias variance identity (Mardia et al., 1979),

$$\mathbb{E}_{\pi^*}[\|Z - \mu\|^2] \quad = \quad \text{Tr}(\text{Cov}_{\pi^*}(Z)) + \|\mathbb{E}_{\pi^*}Z - \mathbb{E}_\pi Z\|^2. \tag{71}$$

Recall the Poincaré Inequality with constant C (Balasubramanian et al., 2022),

$$\text{Var}_{\pi^*}(g(Z)) \quad \le \quad C\int_{\mathbb{R}^n}\|\nabla g(z)\|^2\pi^*(z)dz \tag{72}$$

where $g : \mathbb{R}^n \to \mathbb{R}$ and $g, \|\nabla g(x)\|$ are square-integrable with respect to $\pi^*$. It is a known result where if $\pi^*$ is $\frac{1}{t\delta}$-strongly log-concave, then $\pi^*$ satisfies a Poincaré Inequality with $C = t\delta$. Define a $g : \mathbb{R}^n \to \mathbb{R}$,

$$g(z) = u^T z, \quad u \in \mathbb{R}^n. \tag{73}$$

For measure $\pi^*$ and our defined $g$, (72) gives us,

$$u^\top \text{Var}_{\pi^*}(Z)u \quad \leq \quad t\delta\|u\|^2. \tag{74}$$

Note $\text{Var}_{\pi^*}(Z)$ is a symmetric matrix hence the inequality above implies,

$$\sup_{\|u\|^2=1} u^\top \text{Var}_{\pi^*}(Z)u \leq t\delta \quad \implies \quad \text{Tr}(\text{Var}_{\pi^*}(Z)) \leq nt\delta \tag{75}$$

bounding the quadratic form for all unit vectors bounds all eigenvalues by $t\delta$. Before bounding the bias term, we introduce intermediate reference points

$$z_1^* = \arg\min_z \left\{ f(z) + \frac{1}{2t}\|z-x\|^2 \right\}, \quad z_2^* = \arg\min_z \left\{ 2f(z) + \frac{1}{2t}\|z-x\|^2 \right\}. \tag{76}$$

The factor of 2 on $f(z)$ reflects that $\pi^*$ includes an additional $f(z)$ term from $r(z)$,

$$\pi(dz) \propto \exp\left(-\frac{f(z) + \frac{1}{2t}\|z-x\|^2}{\delta}\right), \quad \pi^*(dz) \propto \exp\left(-\frac{2f(z) + \frac{1}{2t}\|z-x\|^2}{\delta}\right). \tag{77}$$

We decompose the bias as

$$\|\mathbb{E}_{\pi^*}Z - \mathbb{E}_\pi Z\| \quad \leq \quad \|\mathbb{E}_{\pi^*}Z - z_2^*\| + \|z_2^* - z_1^*\| + \|\mathbb{E}_\pi Z - z_1^*\|. \tag{78}$$

By Fermat's rule for the minimizers Theorem 26.2 (Bach, 2013).

$$\frac{1}{t}(x - z_1^*) \quad \in \quad \partial f(z_1^*), \quad \frac{1}{t}(x - z_2^*) \quad \in \quad 2\partial f(z_2^*). \tag{79}$$

Since $f$ is convex and $L$-Lipschitz, all subgradients satisfy $\|g\| \leq L$ for $g \in \partial f(x)$. Therefore,

$$\|z_1^* - x\| \quad \leq \quad tL, \quad \|z_2^* - x\| \quad \leq \quad 2tL. \tag{80}$$

By triangle inequality,

$$\|z_2^* - z_1^*\| \quad \leq \quad \|z_2^* - x\| + \|z_1^* - x\| \quad \leq \quad 3tL, \tag{81}$$

We note that $\pi$ and $\pi^*$ share the same strong-convexity parameter, thus the smoothing error bound $\sqrt{nt\delta}$ applies to both,

$$\|\mathbb{E}_\pi Z - z_1^*\| \quad \leq \quad \sqrt{nt\delta}, \quad \|\mathbb{E}_{\pi^*}Z - z_2^*\| \quad \leq \quad \sqrt{nt\delta}. \tag{82}$$

Putting together (75), (81), and (82), we obtain,

$$\mathbb{E}_{\pi^*}[\|Z - \mu\|^2] \quad \leq \quad nt\delta + (2\sqrt{nt\delta} + 3tL)^2 \quad = \quad M_\star. \tag{83}$$

$\square$

**Lemma B.3** (Bound on $\mathbb{E}_q[r(Z)^2]$). *The expectation of $r(Z)$ under $\pi$ is bounded by $J_\star = \exp(\frac{2}{\delta}L^2 t)$.*

*Proof.* Recall $r(z) = \frac{\pi(z)}{q(z)}$, define

$$c(x) \quad = \quad \frac{(2\pi t\delta)^{n/2}}{C_\delta^x}, \quad w(z) \quad = \quad \exp\left(-\frac{f(z)}{\delta}\right) \tag{84}$$

where our $c(x)$ depends only on $x$ since we have integrated out $z$ from the kernel of $\pi$ and what's left is where our Gaussian Proposal is centered. thus we can decompose,

$$r(z) \quad = \quad c(x)w(z). \tag{85}$$

It is worth taking a look at $\mathbb{E}_q[w(Z)]$,

$$\mathbb{E}_q[w(Z)] \quad = \quad (2\pi\delta t)^{-n/2} \int e^{-f(z)/\delta} e^{-\|z-x\|^2/2t\delta} dz \quad = \quad \frac{C_\delta^x}{(2\pi t\delta)^{n/2}} \quad = \quad \frac{1}{c(x)} \tag{86}$$

thus,

$$\mathbb{E}_\pi[r(Z)] \quad = \quad \mathbb{E}_q[r(Z)^2] \quad = \quad c(x)^2 \mathbb{E}_q[w(Z)^2] \quad = \quad \frac{\mathbb{E}_q[w(Z)^2]}{\mathbb{E}_q[w(Z)]^2}. \tag{87}$$

The Tsirelson-Ibragimov-Sudakov concentration inequality in Theorem 5.5 (Boucheron et al., 2013) bounds $L$-Lipschitz functions of Gaussian random variables. Let $X = (X_1, ... X_n)$ be a vector of n independent standard Normal random variables. Let $h : \mathbb{R}^n \to \mathbb{R}$ denote an $L$-Lipschitz function. Then for all $\lambda \in \mathbb{R}$

$$\log \mathbb{E}[\exp(\lambda(h(X) - \mathbb{E}h(X)))] \quad \leq \quad \frac{\lambda^2}{2} L^2. \tag{88}$$

Let $\Sigma$ be the covariance matrix of $q$ and define our function $h$,

$$h(G) \quad = \quad f(x + \Sigma^{1/2}G), \quad G \quad \sim \quad N(0, I). \tag{89}$$

Two key qualities about $h$ to note is that it is equivalent to function $f(Z)$ when $Z \sim N(x, \delta t I)$ and that $h$ is $\sqrt{\delta t}L$-Lipschitz in $G$. Take the log of (87),

$$\log\left(\frac{\mathbb{E}_q[w(Z)^2]}{\mathbb{E}_q[w(Z)]^2}\right) \quad = \quad \log \mathbb{E}_g[e^{-\frac{2}{\delta}h(G)}] - 2\log \mathbb{E}_g[e^{-\frac{1}{\delta}h(G)}]. \tag{90}$$

By adding and subtracting $\mathbb{E}[h(G)]$ in the exponent, we can upper bound the first term using the TIS inequality with $L = \sqrt{\delta t}$ and $\lambda = -\frac{2}{\delta}$,

$$\log \mathbb{E}_g[e^{-\frac{2}{\delta}h(G)}] \quad \leq \quad -\frac{2}{\delta}\mathbb{E}_g[h(G)] + \frac{2}{\delta}L^2 t. \tag{91}$$

By Jensen's inequality for the concave function log we can lower bound the expectation,

$$2\log \mathbb{E}_g[e^{-\frac{1}{\delta}h(Z)}] \quad \geq \quad 2\mathbb{E}_g[\log e^{-\frac{1}{\delta}h(G)}] \quad = \quad -\frac{2}{\delta}\mathbb{E}_g[h(G)]. \tag{92}$$

Putting together equations (91) and (92) to (90) and exponentiate, we obtain

$$\mathbb{E}_\pi[r(Z)] \quad \leq \quad \exp\left(\frac{2}{\delta}L^2 t\right) = J_\star. \tag{93}$$

$\square$

We can use lemma B.1 and the fact that $\mathbb{E}_\pi[r(Z)] = \mathbb{E}_q[r(Z)^2]$ to rewrite (94),

$$P(\|\hat{\mu}_N - \mu\|^2 > \varepsilon) \quad \leq \quad \frac{4\mathbb{E}_q[r(Z)^2]}{N} + \frac{4\mathbb{E}_q[r(Z)^2]\mathbb{E}_{\pi^*}[\|Z - \mu\|^2]}{N\varepsilon}. \tag{94}$$

Lemma B.3 and B.2 allow us to choose $\varepsilon = \frac{8J_\star M_\star}{\alpha N}$ and for $N \geq \frac{8J_\star}{\alpha}$ for $\alpha \in (0, 1)$ which gives us a nice probabilistic bound,

$$P\left(\|\widehat{\text{prox}}_{tf}^\delta(x) - \text{prox}_{tf}^\delta(x)\| > \sqrt{\frac{8J_\star M_\star}{\alpha N}}\right) \quad \leq \quad \alpha. \tag{95}$$

Reincorporating the smoothing error and triangle inequality in 49, we obtain a high probability error bound for our sampled proximal mapping independent of $x$,

$$P\left(\|\widehat{\text{prox}}_{tf}^\delta(x) - \text{prox}_{tf}(x)\| \leq \sqrt{\frac{8J_\star M_\star}{\alpha N}} + \sqrt{nt\delta}\right) \quad > \quad 1 - \alpha. \tag{96}$$

$\square$

## C. HJ-Prox-based PPM Convergence

For completeness and ease of presentation, we restate the theorem.

**Proof of Thm. 3.5.** *Let $f, g$ be proper, LSC, convex and L-Lipschitz. Consider the HJ-Prox-based PPM iteration given by*

$$x_{k+1} \;=\; \widehat{\text{prox}}_{t_k(f+g)}^{\delta_k}(x_k), \tag{97}$$

*with parameters satisfying the conditions of Assumption 3.4. Then $x_k$ converges almost surely to a minimizer of $f + g$.*

*Proof.*

**Lemma C.1** (PPM Diminishing Stepsizes). *Consider the nonempty solution set for LSC and convex $f$, $S = \arg\min f$. Define the PPM iterations generated by $x_{k+1} = \text{prox}_{t_k f}(x_k)$ with step sizes $t_k = 1/(k+1)$. Then, every cluster point of the sequence $(x_k)_{k \geq 0}$ is an optimal solution in $S$.*

*Proof.* For a fixed $t > 0$, let $y = \text{prox}_{tf}(x)$. The optimality condition $\frac{x-y}{t} \in \partial f(y)$ and convexity of $f$ yields

$$f(u) \;\geq\; f(y) + \left\langle \frac{x-y}{t}, u - y \right\rangle \tag{98}$$

for all $u \in \mathbb{R}^n$. Using the identity $2\langle a, b \rangle = \|a\|^2 + \|b\|^2 - \|a-b\|^2$ with $a = x - y$ and $b = u - y$, we obtain the standard three-point inequality,

$$2t(f(y) - f(u)) \;\leq\; \|x - u\|^2 - \|y - u\|^2 - \|x - y\|^2. \tag{99}$$

Setting $x = x_k$, $y = x_{k+1}$, and $u \in S$ (so that $f(u) = f^*$) gives

$$2t_k(f(x_{k+1}) - f^*) \;\leq\; \|x_k - u\|^2 - \|x_{k+1} - u\|^2 - \|x_{k+1} - x_k\|^2, \tag{100}$$

which implies $\|x_{k+1} - u\| \leq \|x_k - u\|$. Thus, the sequence $(x_k)$ is Fejér monotone with respect to $S$ and is therefore bounded and at least one cluster point exists. Summing the inequality over $k$ yields

$$\sum_{k=0}^{\infty} t_k(f(x_{k+1}) - f^*) \;\leq\; \frac{1}{2}\|x_0 - u\|^2 < \infty. \tag{101}$$

Setting $u = x_k$ in (99) shows that the function value is non-increasing

$$f(x_{k+1}) \;\leq\; f(x_k) - \frac{1}{2t_k}\|x_{k+1} - x_k\|^2 \;\leq\; f(x_k). \tag{102}$$

Since $f(x_k)$ is decreasing and bounded below by $f^*$, it converges to some limit $L \geq f^*$. We claim $L = f^*$. If $L > f^*$, then $f(x_{k+1}) - f^* \geq L - f^* > 0$. For sufficiently large $k$, this strictly positive gap combined with the divergent step size assumption $\sum t_k = \infty$ (specifically $t_k = 1/(k+1)$) would imply

$$\sum_{k=0}^{\infty} t_k(f(x_{k+1}) - f^*) \;\geq\; (L - f^*) \sum_{k=0}^{\infty} t_k \;=\; \infty, \tag{103}$$

which contradicts the summability condition (101). Therefore, we must have $\lim_{k \to \infty} f(x_k) = f^*$. Finally, let $\bar{x}$ be any cluster point of the sequence, with a subsequence $x_{k_n} \to \bar{x}$. By the lsc of $f$,

$$f(\bar{x}) \;\leq\; \liminf_{n \to \infty} f(x_{k_n}) \;=\; \lim_{k \to \infty} f(x_k) \;=\; f^*. \tag{104}$$

Since $f(\bar{x})$ cannot be less than the global minimum $f^*$, we conclude $f(\bar{x}) = f^*$, implying $\bar{x} \in S$. $\qquad \square$

The PPM iterates are computed by applying the mapping $T_k(x) = \text{prox}_{t_k(f+g)}(x)$. By Lemma C.1 we show that every cluster point lies in $\bigcap_{k \geq 0} \text{Fix } T_k$ and it is known that $T_k$ is an averaged operator when $(f + g)$ is convex.

The HJ-PPM iterates can be written as

$$\hat{x}_{k+1} \quad = \quad \widehat{\text{prox}}_{t_k(f+g)}^{\delta_k}(\hat{x}_k) \quad = \quad T_k(\hat{x}_k) + \varepsilon_k, \tag{105}$$

where

$$\varepsilon_k \quad = \quad \widehat{\text{prox}}_{t_k(f+g)}^{\delta_k}(\hat{x}_k) - \text{prox}_{t_k(f+g)}(\hat{x}_k). \tag{106}$$

From Theorem 3.3, we uniformly bound,

$$P\left( \|\varepsilon_k\| > \sqrt{\frac{8 J_\star M_\star}{\alpha_k N_k}} + \sqrt{n t_k \delta_k} \right) \quad \leq \quad \alpha_k. \tag{107}$$

Under assumption 3.4,

$$\sum_{k=0}^{\infty} \alpha_k < \infty \quad \text{and} \quad \sum_{k=0}^{\infty} \sqrt{\frac{8 J_\star M_\star}{\alpha_k N_k}} + \sqrt{n t_k \delta_k} < \infty, \tag{108}$$

thus we can invoke Borel-Cantelli,

$$\sum_{k=0}^{\infty} \|\varepsilon_k\| \quad < \quad \infty \quad \text{almost surely.} \tag{109}$$

We have verified all conditions of Theorem 3.1: each $T_k$ is an averaged operator, the fixed point set $S = \bigcap_{k \geq 0} \text{Fix } T_k$ is nonempty, $\sum_{k=0}^{\infty} \|\varepsilon_k\| < \infty$ a.s., and by Lemma C.1, every cluster point of the perturbed sequence lies in $S$. Therefore, by Theorem 3.1 applied pathwise, the sequence $\{\hat{x}_k\}_{k \geq 0}$ converges almost surely to some $x^* \in S = \arg\min(f + g)$. $\qquad \square$

## D. HJ-Prox-based PGD Convergence

For completeness and ease of presentation, we restate the theorem.

**Proof of Thm. 3.6.**  *Let $f, g$ be proper, LSC, convex and $L$-Lipschitz, with $f$ additionally $L'$-smooth. Consider the HJ-Prox-based PGD iteration given by*

$$x_{k+1} \quad = \quad \widehat{\text{prox}}_{t_k g}^{\delta_k}(x_k - t_k \nabla f(x_k)), \tag{110}$$

*with step size $0 < t_k < 1/L'$ and parameters satisfying the conditions of Assumption 3.4. Then $x_k$ converges almost surely to a minimizer of $f + g$.*

*Proof.* For appropriately chosen step-size $t$, the PGD algorithm map is averaged and its fixed points coincide with the global minimizers of $f$ (as shown in the Lemma below).

**Lemma D.1** (PGD Diminishing Stepsizes). *Consider $F = f + g$ where $f$ is convex with $L$-Lipschitz continuous gradient, $g$ is proper, convex and lsc, and the solution set $S = \arg\min F$ is nonempty. The proximal gradient iterations*

$$x_{k+1} \quad = \quad \text{prox}_{t_k g}(x_k - t_k \nabla f(x_k)) \tag{111}$$

*with $t_k \leq 1/L$ and $\sum_{k=0}^{\infty} t_k = \infty$ then every cluster point of $(x_k)_{k \geq 0}$ lies in $S$.*

*Proof.* Let $u \in \mathbb{R}^n$. Using the proximal optimality condition for $g$ at $x_{k+1} = \text{prox}_{t_k g}(x_k - t_k \nabla f(x_k))$, convexity of $f$, and the $L$-smoothness of $f$, we obtain the standard proximal-gradient three-point inequality

$$2 t_k (F(x_{k+1}) - F(u)) \quad \leq \quad \|x_k - u\|^2 - \|x_{k+1} - u\|^2 - (1 - L t_k)\|x_{k+1} - x_k\|^2. \tag{112}$$

Taking $u \in S$ (so $F(u) = F^*$) and using $t_k \le 1/L$ gives

$$2t_k(F(x_{k+1}) - F^*) \quad \le \quad \|x_k - u\|^2 - \|x_{k+1} - u\|^2, \tag{113}$$

hence $\|x_{k+1} - u\| \le \|x_k - u\|$. Thus $(x_k)$ is Fejér monotone with respect to $S$, hence bounded with at least one cluster point. Summing (112) over $k$ and dropping the nonnegative term $(1 - Lt_k)\|x_{k+1} - x_k\|^2$ yields

$$\sum_{k=0}^{\infty} t_k(F(x_{k+1}) - F^*) \quad \le \quad \frac{1}{2}\|x_0 - u\|^2 < \infty. \tag{114}$$

We take $u = x_k$ in (112)

$$2t_k(F(x_{k+1}) - F(x_k)) \quad \le \quad -(1 - Lt_k)\|x_{k+1} - x_k\|^2 \le 0, \tag{115}$$

so $F(x_k)$ is non-increasing, hence converges to some $L \ge F^*$. If $L > F^*$, then $F(x_{k+1}) - F^* \ge L - F^* > 0$ for all sufficiently large $k$, which combined with $\sum_{k=0}^{\infty} t_k = \infty$ implies

$$\sum_{k=0}^{\infty} t_k(F(x_{k+1}) - F^*) \quad \ge \quad (L - F^*)\sum_{k=0}^{\infty} t_k \quad = \quad \infty, \tag{116}$$

contradicting (114). Thus $\lim_{k\to\infty} F(x_k) = F^*$.

Finally, let $\bar{x}$ be any cluster point with $x_{k_n} \to \bar{x}$. Since $f$ is continuous and $g$ is lsc, $F = f + g$ is lsc, hence

$$F(\bar{x}) \quad \le \quad \liminf_{n\to\infty} F(x_{k_n}) \quad = \quad F^*, \tag{117}$$

which implies $\bar{x} \in S$. $\qquad\square$

**Lemma D.2** (Averagedness and Fixed Points of PGD). *Let $0 < t < \frac{2}{L}$ and define, for $x \in \mathbb{R}^n$*

$$T(x) \quad = \quad \mathrm{prox}_{tg}\big(x - t\nabla f(x)\big). \tag{118}$$

*Then $T$ is an averaged operator, and its fixed points Fix$(T)$ coincide with $f + g$'s global minimizers $X^*$ (Parikh & Boyd, 2014) section 4.2.*

The PGD iterates are computed by applying the mapping $T_k(x) = \mathrm{prox}_{t_k g}(x - t_k\nabla f(x))$. By Lemma D.2 and D.1, all $T_k$ is an averaged operator and $x_k \to x^* \in \bigcap_{k \ge 0} \mathrm{Fix}\, T_k = X^*$.

The HJ-PGD iterates can be written as

$$\hat{x}_{k+1} \quad = \quad \widehat{\mathrm{prox}}_{tg}^{\delta_k}(\hat{x}_k - t_k\nabla f(\hat{x}_k)) \quad = \quad T_k(\hat{x}_k) + \varepsilon_k, \tag{119}$$

where

$$\varepsilon_k \quad = \quad \widehat{\mathrm{prox}}_{t_k g}^{\delta_k}(\hat{x}_k - t_k\nabla f(\hat{x}_k)) - \mathrm{prox}_{t_k g}(\hat{x}_k - t_k\nabla f(\hat{x}_k)). \tag{120}$$

From Theorem 3.3, we have the uniform bound

$$P\left(\|\varepsilon_k\| > \sqrt{\frac{8J_\star M_\star}{\alpha_k N_k}} + \sqrt{nt_k\delta_k}\right) \quad \le \quad \alpha_k. \tag{121}$$

Under Assumption 3.4, we have

$$\sum_{k=0}^{\infty} \alpha_k < \infty \quad \text{and} \quad \sum_{k=0}^{\infty}\left(\sqrt{\frac{8J_\star M_\star}{\alpha_k N_k}} + \sqrt{nt_k\delta_k}\right) < \infty. \tag{122}$$

Thus, by Borel-Cantelli,

$$\sum_{k=0}^{\infty} \|\varepsilon_k\| \quad < \quad \infty \quad \text{almost surely.} \tag{123}$$

We have now verified all conditions of Theorem 3.1. By Lemma D.2, $T$ is an averaged operator for step sizes $0 < t_k < 2/L$, and the fixed point set $S = \operatorname{Fix} T = \arg\min(f + g)$ is nonempty by assumption. $\sum_{k=0}^{\infty} \|\varepsilon_k\| < \infty$ almost surely and by Lemma D.1, every cluster point of the perturbed sequence lies in $S$. Therefore, by Theorem 3.1 applied pathwise, the sequence $\{\hat{x}_k\}_{k \geq 0}$ converges almost surely to some $x^* \in S = \arg\min(f + g)$. $\qquad \square$

## E. HJ-Prox-based DRS Convergence

We restate the statement of the theorem for readability.

**Proof of Thm. 3.8.** *Let $f, g$ be proper, convex, LSC, and L-Lipschitz. Consider the HJ-Prox–based DRS iteration given by*

$$\begin{aligned}
x_{k+1/2} &= \widehat{\operatorname{prox}}_{tf}^{\delta_k}(z_k), \\
x_{k+1} &= \widehat{\operatorname{prox}}_{tg}^{\delta_k}(2x_{k+1/2} - z_k), \\
z_{k+1} &= z_k + x_{k+1} - x_{k+1/2},
\end{aligned} \tag{124}$$

*with parameters satisfying the conditions of Assumption 3.7. Then $x_k$ converges almost surely to a minimizer of $f + g$.*

*Proof.* The DRS algorithm map is averaged and its fixed points coincide with the global minimizers of $f + g$.

**Lemma E.1** (Averagedness and Fixed Points of DRS). *Let $t > 0$ and define, for $z \in \mathbb{R}^n$*

$$T(z) \quad = \quad z + \operatorname{prox}_{tg}(2 \operatorname{prox}_{tf}(z) - z) - \operatorname{prox}_{tf}(z). \tag{125}$$

*Note this is the fixed point operator for the dual variable in the DRS algorithm. Then $T$ is firmly nonexpansive (hence averaged), and*

$$Fix(T) \quad = \quad \{z : \operatorname{prox}_{tf}(z) \in Z^*\}. \tag{126}$$

*(Lions & Mercier, 1979) (remark 5).*

By Lemma E.1, $z_k \to z^*$ and $\operatorname{prox}(z^*) = x^* \in X^*$. We can express the HJ-DRS update in terms of the DRS algorithm map $T$ (125).

$$\hat{z}_{k+1} \quad = \quad T(\hat{z}_k) + \varepsilon_k, \tag{127}$$

where

$$\begin{aligned}
\varepsilon_k &= \operatorname{prox}_{tg}(w_k + 2\kappa_k) - \operatorname{prox}_{tg}(w_k) + \zeta_k - \kappa_k, \tag{128} \\
w_k &= 2 \operatorname{prox}_{tf}(\hat{z}_k) - \hat{z}_k, \tag{129}
\end{aligned}$$

and

$$\begin{aligned}
\zeta_k &= \widehat{\operatorname{prox}}_{tg}^{\delta_k}(\hat{z}_k) - \operatorname{prox}_{tg}(\hat{z}_k) \tag{130} \\
\kappa_k &= \widehat{\operatorname{prox}}_{tf}^{\delta_k}(\hat{z}_k) - \operatorname{prox}_{tf}(\hat{z}_k). \tag{131}
\end{aligned}$$

We have the following bound

$$\|\varepsilon_k\| \quad \leq \quad \|w_k + 2\kappa_k - w_k\| + \|\zeta_k\| + \|\kappa_k\| \quad = \quad 3\|\kappa_k\| + \|\zeta_k\|, \tag{132}$$

which follows from the triangle inequality and the fact that proximal mappings are nonexpansive. From Theorem 3.3, we have the uniform bound

$$P\left(\|\zeta_k\| > \sqrt{\frac{8J_\star M_\star}{\alpha_k N_k}} + \sqrt{nt_k\delta_k}\right) \leq \alpha_k, \tag{133}$$

$$P\left(\|\kappa_k\| > \sqrt{\frac{8J_\star M_\star}{\alpha_k N_k}} + \sqrt{nt_k\delta_k}\right) \leq \alpha_k. \tag{134}$$

Under Assumption 3.7, we have

$$\sum_{k=0}^{\infty} \alpha_k < \infty \quad \text{and} \quad \sum_{k=0}^{\infty}\left(\sqrt{\frac{8J_\star M_\star}{\alpha_k N_k}} + \sqrt{nt\delta_k}\right) < \infty. \tag{135}$$

Thus, by Borel-Cantelli,

$$\sum_{k=0}^{\infty} \|\zeta_k\| < \infty \quad \text{and} \quad \sum_{k=0}^{\infty} \|\kappa_k\| < \infty \quad \text{almost surely.} \tag{136}$$

By equation (132),

$$\sum_{k=0}^{\infty} \|\varepsilon_k\| < \infty \quad \text{almost surely.} \tag{137}$$

We have verified all conditions of Theorem 3.1. By Lemma E.1, $T$ is firmly nonexpansive with nonempty fixed point set $S = \text{Fix}(T) = \{z : \text{prox}_{tf}(z) \in X^*\}$, where $X^* = \arg\min(f+g)$. We have shown $\sum_{k=0}^{\infty} \|\varepsilon_k\| < \infty$ almost surely. Since the step size $t$ is fixed, we have a constant operator $T$ across all iterations, and the demiclosedness of $T - \text{Id}$ at zero ensures every cluster point lies in $S$ (Combettes, 2001). Therefore, by Theorem 3.1, $\hat{z}_k \to z^* \in S$ almost surely. Since proximal maps are continuous, $\hat{x}_k = \widehat{\text{prox}}_{tf}^{\delta_k}(\hat{z}_k) \to \text{prox}_{tf}(z^*) = x^* \in X^*$ almost surely. $\qquad\square$

## F. HJ-Prox-based DYS Convergence

For completeness and ease of presentation, we restate the theorem.

**Proof of Thm. 3.9.** *For DYS, consider $f + g + h$. Let $f, g, h$ be proper, LSC, convex and L-Lipschitz, with $h$ additionally $L'$-smooth. Consider the HJ-Prox–based DYS algorithm given by*

$$\begin{aligned}
y_{k+1} &= \widehat{\text{prox}}_{tf}^{\delta_k}(x_k), \\
z_{k+1} &= \widehat{\text{prox}}_{tg}^{\delta_k}\left(2y_{k+1} - x_k - t\nabla h(y_{k+1})\right) \\
x_{k+1} &= x_k + z_{k+1} - y_{k+1}
\end{aligned} \tag{138}$$

*with parameters satisfying the conditions of Assumption 3.7 and $0 < t < 2/L'$. Then $x_k$ converges almost surely to a minimizer of $f + g + h$.*

*Proof.* For appropriately chosen step-size $t$, the DYS algorithm map is averaged and its fixed points coincide with the global minimizers of $f + g + h$.

**Lemma F.1** (Averagedness and Fixed Points of DYS). *Let $t > 0$ and define, for $z \in \mathbb{R}^n$,*

$$T(z) = z - \text{prox}_{tf}(z) + \text{prox}_{tg}\left(2\text{prox}_{tf}(z) - z - t\nabla h(\text{prox}_{tf}(z))\right). \tag{139}$$

*Note this is the fixed point operator for the DYS algorithm and its fixed points Fix(T) coincide with global minimizers $X^*$. $T$ is firmly nonexpansive (hence averaged), and*

$$Fix(T) = \{z : x \in X^\star\}, \tag{140}$$

*(Davis & Yin, 2017)(Theorem 3.1).*

By Lemma F.1, $z_k \to z^\star$ and $z^* \in X^*$. We can express the HJ-DYS update in terms of DYS algorithm map $T$ (139).

$$\hat{z}_{k+1} = T(\hat{z}_k) + \varepsilon_k, \tag{141}$$

where

$$\varepsilon_k = \text{prox}_{tg}\big(S_t(z_k) + d_k\big) - \text{prox}_{tg}\big(S_t(z_k)\big) + \zeta_k - \kappa_k \tag{142}$$

$$S_t(z_k) = 2\,\text{prox}_{tf}(z_k) - z_k - t\nabla h\big(\text{prox}_{tf}(z_k)\big) \tag{143}$$

$$d_k = 2\kappa_k - t[\nabla h\big(\text{prox}_{tf}(z_k) + \kappa_k\big) - \nabla h\big(\text{prox}_{tg}(z_k)\big)] \tag{144}$$

and

$$\zeta_k = \widehat{\text{prox}}_{tg}^{\delta_k}(\hat{z}_k) - \text{prox}_{tg}(\hat{z}_k) \tag{145}$$

$$\kappa_k = \widehat{\text{prox}}_{tf}^{\delta_k}(\hat{z}_k) - \text{prox}_{tf}(\hat{z}_k). \tag{146}$$

We have the following bound

$$\|\varepsilon_k\| \leq (1 + tL)\|\kappa_k\| + \|\zeta_k\|, \tag{147}$$

which follows from the triangle inequality, $L$-smoothness of $h$, and the fact that proximal mappings are nonexpansive. From Theorem 3.3, we have the uniform bound

$$P\left(\|\zeta_k\| > \sqrt{\frac{8J_\star M_\star}{\alpha_k N_k}} + \sqrt{nt_k\delta_k}\right) \leq \alpha_k, \tag{148}$$

$$P\left(\|\kappa_k\| > \sqrt{\frac{8J_\star M_\star}{\alpha_k N_k}} + \sqrt{nt_k\delta_k}\right) \leq \alpha_k. \tag{149}$$

Under Assumption 3.7, we have

$$\sum_{k=0}^{\infty} \alpha_k < \infty \quad \text{and} \quad \sum_{k=0}^{\infty}\left(\sqrt{\frac{8J_\star M_\star}{\alpha_k N_k}} + \sqrt{nt\delta_k}\right) < \infty. \tag{150}$$

Thus, by Borel-Cantelli,

$$\sum_{k=0}^{\infty} \|\zeta_k\| < \infty \quad \text{and} \quad \sum_{k=0}^{\infty} \|\kappa_k\| < \infty \quad \text{almost surely.} \tag{151}$$

By equation (147),

$$\sum_{k=0}^{\infty} \|\varepsilon_k\| < \infty \quad \text{almost surely.} \tag{152}$$

We have verified all conditions of Theorem 3.1. By Lemma F.1, $T$ is firmly nonexpansive with nonempty fixed point set $S = \text{Fix}(T) = \{z : z \in X^*\}$, where $X^* = \arg\min(f + g + h)$. We have shown $\sum_{k=0}^{\infty} \|\varepsilon_k\| < \infty$ almost surely. Since the step size $t$ is fixed, we have a constant operator $T$ across all iterations, and the demiclosedness of $T - \text{Id}$ at zero ensures every cluster point lies in $S$ (Combettes, 2001). Therefore, by Theorem 3.1, $\hat{z}_k \to z^* \in S = X^*$ almost surely. □

## G. HJ-Prox-based PDHG Convergence

For completeness and ease of presentation, we restate the theorem.

**Proof of Thm. 3.10.** *Let $f, g$ be proper, convex, and LSC. Consider the HJ-Prox–based PDHG algorithm given by*

$$
\begin{aligned}
y_{k+1} &= \widehat{\text{prox}}_{\sigma g^*}^{\delta_k}(y_k + \sigma A x_k), \\
x_{k+1} &= \widehat{\text{prox}}_{\tau f}^{\delta_k}(x_k - \tau A^\top y_{k+1}),
\end{aligned}
\tag{153}
$$

*with parameters $\tau, \sigma > 0$ satisfying $\tau\sigma\|A\|^2 < 1$ and the conditions of t in Assumption 3.7, where $g^*$ denotes the Fenchel conjugate of g. Then $x^k$ converges almost surely to a minimizer of $f(x) + g(Ax)$.*

*Proof.* For appropriately chosen $\tau, \sigma$ the PDHG algorithm map is averaged and its fixed points corresponding to $x_k$ updates coincide with the global minimizers of $f(x) + g(Ax)$.

**Lemma G.1** (Averagedness and Fixed Points of PDHG). *Let $\tau, \sigma > 0$ satisfying $\tau\sigma\|A\|^2 < 1$ and define, for $z \in \mathbb{R}^n$ and $w \in \mathbb{R}^m$*

$$
T(z, w) = \begin{bmatrix} \text{prox}_{\tau f}\big(z - \tau A^\top \text{prox}_{\sigma g^*}(w + \sigma Az)\big) \\ \text{prox}_{\sigma g^*}(w + \sigma Az) \end{bmatrix}.
\tag{154}
$$

*Let $V = \text{diag}(\frac{1}{\tau}I_n, \frac{1}{\sigma}I_m)$. On a product space with a weighted inner product $\langle(x, y), (x', y')\rangle_V = \frac{1}{\tau}\langle x, x'\rangle + \frac{1}{\sigma}\langle y, y'\rangle$, the map T is an averaged operator. Note this is the fixed point operator for the PDHG algorithm and its fixed points Fix(T) coincide with the set of primal-dual KKT saddle points for $f(x) + g(Ax)$, where the primal point coincides with the global minimizers $X^*$. T is firmly nonexpansive (hence averaged), and*

$$
Fix(T) = \{(z^*, w^*) : z^* \in X^\star\}
\tag{155}
$$

*(Chambolle & Pock, 2011)(Algorithm 1, Thm. 1) (Fercoq, 2022)(Lemma 2).*

By Lemma G.1, $z_k \to z^*$ and $z^* \in X^*$. We can express the HJ-PDHG update in terms of PDHG algorithm map $T$ (154).

$$
(\hat{z}_{k+1}, \hat{w}_{k+1}) = T(\hat{z}_k, \hat{w}_k) + \varepsilon_k
\tag{156}
$$

where

$$
\varepsilon_k = \begin{bmatrix} \text{prox}_{\tau f}(u_k - \tau A^\top \zeta_k) - \text{prox}_{\tau f}(u_k) + \kappa_k \\ \zeta_k \end{bmatrix}
\tag{157}
$$

$$
u_k = \hat{z}_k - \tau A^\top \text{prox}_{\sigma g^*}(\hat{w}_k + \sigma A\hat{z}_k),
\tag{158}
$$

and

$$
\zeta_k = \text{prox}_{\sigma g^*}^{\delta_k}(\hat{w}_k + \sigma A\hat{z}_k) - \text{prox}_{\sigma g^*}(\hat{w}_k + \sigma A\hat{z}_k)
\tag{159}
$$

$$
\kappa_k = \text{prox}_{\tau f}^{\delta_k}(u_k - \tau A^\top \zeta_k) - \text{prox}_{\tau f}(u_k - \tau A^\top \zeta_k)
\tag{160}
$$

In the weighted norm $\|(z, w)\|_V^2 = \frac{1}{\tau}\|z\|^2 + \frac{1}{\sigma}\|w\|^2$, we have the following bound

$$
\|\varepsilon_k\|_V^2 \leq \big(2\tau\|A\|_{\text{op}}^2 + \frac{1}{\sigma}\big)\|\zeta_k\|^2 + \frac{2}{\tau}\|\kappa_k\|^2
\tag{161}
$$

which follows from the fact that proximal mappings are nonexpansive and from $\|A^\top\|_{\text{op}} = \|A\|_{\text{op}}$.

From Theorem 3.3, we have the uniform bound

$$
P\left(\|\zeta_k\| > \sqrt{\frac{8J_\star M_\star}{\alpha_k N_k}} + \sqrt{nt_k\delta_k}\right) \leq \alpha_k,
\tag{162}
$$

$$
P\left(\|\kappa_k\| > \sqrt{\frac{8J_\star M_\star}{\alpha_k N_k}} + \sqrt{nt_k\delta_k}\right) \leq \alpha_k.
\tag{163}
$$

Under Assumption 3.7, we have

$$\sum_{k=0}^{\infty} \alpha_k < \infty \quad \text{and} \quad \sum_{k=0}^{\infty} \left( \sqrt{\frac{8 J_\star M_\star}{\alpha_k N_k}} + \sqrt{n t \delta_k} \right) < \infty. \tag{164}$$

Thus, by Borel-Cantelli,

$$\sum_{k=0}^{\infty} \|\zeta_k\| \quad < \quad \infty \quad \text{and} \quad \sum_{k=0}^{\infty} \|\kappa_k\| \quad < \quad \infty \quad \text{almost surely.} \tag{165}$$

By equation (161),

$$\sum_{k=0}^{\infty} \|\varepsilon_k\|_V \quad < \quad \infty \quad \text{almost surely.} \tag{166}$$

We have verified all conditions of Theorem 3.1 in the weighted metric space. By Lemma G.1, $T$ is firmly nonexpansive (hence averaged), and its fixed point set $S = \text{Fix}(T) = \{(z^*, w^*) : z^* \in X^*\}$. We have shown $\sum_{k=0}^{\infty} \|\varepsilon_k\|_V < \infty$ almost surely. Since the step sizes $\tau, \sigma$ are fixed, we have a constant operator $T$ across all iterations, so $\bigcap_{k \geq 0} \text{Fix}(T_k) = \text{Fix}(T) = S$. The closedness condition follows from the demiclosedness of $T - \text{Id}$ at zero for firmly nonexpansive operators in the weighted metric (Combettes, 2001). Therefore, by Theorem 3.1, $(\hat{z}_k, \hat{w}_k) \to (z^*, w^*) \in S$ almost surely, where $z^* \in X^* = \arg\min(f(x) + g(Ax))$ is a global minimizer. $\qquad \square$

# H. Experiment Details

HJ-Prox and analytical counterparts run through all iterations. Every experiment simulates a ground truth structure with added noise and blur depending on problem setup. All parameters and step sizes are matched between HJ-Prox and the analytical counterparts to ensure a fair comparison. The HJ-Prox $\delta$ sequence follows a schedule

$$\delta_k \quad = \quad \mathcal{O}\left( \frac{1}{k^{2+p}} \right), \quad p > 0 \tag{167}$$

where $k$ denotes the iteration number. The defined schedule decays strictly faster than $1/k^2$ satisfying conditions used in Theorem 3.1. For our experiments, we set $p = 0.00001$. We note that the convergence behavior is robust to substantial deviations from the $\delta_k = 1/k^{2+p}$ guideline. In our scope of experiments, power-law schedules in the range $\delta_k \sim 1/k$ to $\delta_k \sim 1/k^2$ produce the best solution quality. A subtle but important point worth emphasizing is that the confidence sequence $\alpha_k$ is a proof-level device used in the Borel-Cantelli argument, *not an algorithmic hyperparameter*. Because the theoretically sufficient sample size scales as $1/\alpha_k$ in Theorem 3.3, choosing $\alpha_k$ to decay too aggressively is doubly wasteful: it has no effect on the actual iterates, and it inflates the sample budget that the theory demands. Theoretically, the favored approach is therefore the slowest summable power-law schedules for both $\delta_k$ and $\alpha_k$ where we are fast enough to control the cumulative approximation error but no faster than necessary.

## H.1. PGD: LASSO Regression

We solve the classic LASSO regression problem using PGD. The simulation setup involves a design matrix $X \in \mathbb{R}^{250 \times 500}$ with 250 observations and 500 predictors. The true coefficients $\beta$ are set such that $\beta^{400:410} = 1$ and all others are zero. The objective function is written as,

$$\arg\min_{\beta} \frac{1}{2} \|X\beta - y\|_2^2 + \lambda \|\beta\|_1 \tag{168}$$

$$X \in \mathbb{R}^{250 \times 500}, \quad \beta \in \mathbb{R}^{500}, \quad y \in \mathbb{R}^{250}.$$

The analytical PGD baseline performs a gradient step on the least-squares term followed by the exact soft thresholding.

## H.2. DRS: Multitask Learning

Multitask learning learns predictive models for multiple related response variables by sharing information across tasks to enhance performance. We solve this problem using Douglas-Rachford splitting, employing HJ-Prox in place of analytical updates. We group the quadratic loss with the nuclear norm regularizer to form one function and the row and column group LASSO terms to form the other. Both resulting functions are non-smooth, requiring HJ-Prox for their proximal mappings. The simulation setup involves $n = 50$ observations, $p = 30$ predictors, and $q = 9$ tasks. The objective function is written as,

$$\arg\min_{\beta} \frac{1}{2} \|XB - Y\|_F^2 \; + \; \lambda_1 \|B\|_* \; + \; \lambda_2 \sum_i \|b_{i,\cdot}\|_2 \; + \; \lambda_3 \sum_j \|b_{\cdot,j}\|_2 \tag{169}$$

$$X \in \mathbb{R}^{50 \times 30}, \quad B \in \mathbb{R}^{30 \times 9}, \quad Y \in \mathbb{R}^{50 \times 9}.$$

The analytical counterpart for Douglas Rachford Splitting utilizes singular value soft thresholding for the nuclear norm and group soft thresholding for the row and column penalties. These regularizers are integrated with fast iterative soft thresholding (FISTA) to handle the data fidelity term with nuclear norm regularization and Dykstra's algorithm to handle the sum of row and column group LASSO penalties.

## H.3. DRS: Trend Filtering

Trend filtering is commonly used in signal processing to promote piecewise smoothness in the solution (Tibshirani, 2014). We apply it to recover a Doppler signal with length $n = 256$ using a third-order differencing matrix $D$. We solve this problem with DRS, comparing two implementation strategies: an exact method using product-space reformulation motivated by (Tibshirani & Taylor, 2011), and an approximate method using HJ-Prox. The objective function is written as,

$$\arg\min_{\beta} \frac{1}{2} \|\beta - y\|^2 + \lambda \|D\beta\|_1 \tag{170}$$

$$\beta \in \mathbb{R}^{256}, \quad y \in \mathbb{R}^{256}, \quad D \in \mathbb{R}^{253 \times 256}.$$

The proximal operator of $\lambda \|D\beta\|_1$ has no closed-form solution for general linear operators $D$. The analytical counterpart addresses this by reformulating the problem in a product space with auxiliary variable $w = D\beta$, yielding separable proximal operators (weighted averaging and soft thresholding) at the cost of inverting terms including $D^\top D$ at each iteration. In contrast, our HJ-Prox variant directly approximates the intractable proximal operator through Monte Carlo sampling.

## H.4. DYS: Sparse Group LASSO

The sparse group LASSO promotes group-level sparsity while allowing individual variable selection within groups, which is useful when certain groups are relevant but contain unnecessary variables. We solve this problem using DYS, employing HJ-Prox for the proximal operators of the non-smooth regularizers. The simulation setup involves $n = 300$ observations with $G = 6$ groups, each having 10 predictors. The objective function is written as,

$$\arg\min_{\beta} \frac{1}{2} \|X\beta - y\|_2^2 \; + \; \lambda_1 \sum_{g=1}^{6} \|\beta_g\|_2 \; + \; \lambda_2 \|\beta\|_1 \tag{171}$$

$$X \in \mathbb{R}^{300 \times 60}, \quad \beta \in \mathbb{R}^{60}, \quad y \in \mathbb{R}^{300}.$$

The analytical counterpart for DYS solves the sparse group LASSO by using soft thresholding for the $\ell_1$ penalty and group soft thresholding for the group $\ell_2$ penalty.

## H.5. PDHG: Total Variation

Lastly, we implement PDHG method to solve the isotropic total variation regularized least-squares problem. We apply the proximal operator for the data fidelity term via its closed-form update and employ our HJ-based proximal operator for the total variation penalty. For this experiment, we recover a smoothed 64 x 64 black and white image from a noisy and blurred image $y$. The objective function is written as,

$$\arg\min_{\beta} \frac{1}{2} \|X\beta - y\|_F^2 + \lambda \mathrm{TV}(\beta) \tag{172}$$

$$\beta \in \mathbb{R}^{64 \times 64}, \quad y \in \mathbb{R}^{64 \times 64}.$$

The (slightly smoothed) isotropic TV we use to evaluate the objective is

$$\mathrm{TV}(\beta) \;=\; \sum_{i=1}^{64} \sum_{j=1}^{64} \sqrt{(\nabla_x \beta)_{i,j}^2 + (\nabla_y \beta)_{i,j}^2}. \tag{173}$$

The analytical counterpart for PDHG algorithm updates dual variables using closed-form scaling for data fidelity and clamping (for $\ell_2$ projection of TV dual), and primal variables using Fast Fourier transform convolution and divergence via finite differences.

## H.6. DYS: Non-negative LASSO

The non-negative LASSO extends standard LASSO regression by enforcing non-negativity constraints on the coefficients, which is appropriate when the underlying relationship is known to be monotonic or when negative coefficients lack physical interpretation. We solve this problem using DYS, employing HJ-Prox for the proximal operators of the non-smooth regularizer and constraint. The simulation setup involves 250 observations and $p = 500$, with 50 coefficients truly nonzero and positive. We use a fixed $\delta$ here to truly showcase the errors. The objective function is written as,

$$\arg\min_{\beta} \frac{1}{2} \|X\beta - y\|_2^2 + \lambda \|\beta\|_1 + I_{\mathbb{R}_+^p}(\beta) \tag{174}$$

$$X \in \mathbb{R}^{250 \times 500}, \quad \beta \in \mathbb{R}^{500}, \quad y \in \mathbb{R}^{250}.$$

where $I_{\mathbb{R}_+^p}(\beta)$ is the indicator function for the non-negative orthant, equal to 0 when all elements of $\beta$ non-negative and $+\infty$ otherwise. This formulation incorporates the non-negativity constraint directly into the objective function.

The analytical counterpart for DYS solves the non-negative LASSO using three separable proximal operators: soft thresholding for the $\ell_1$ penalty, projection onto the non-negative orthant (element-wise maximum with zero), and the resolvent of the gradient for the least-squares term.

## H.7. DYS: Overlapping Group LASSO

We consider the following overlapping group LASSO problem,

$$\arg\min_{\beta} \frac{1}{2} \|X\beta - y\|_2^2 \;+\; \lambda_1 \sum_{g=1}^{298} w_g \|\beta_g\|_2 \;+\; \lambda_2 \|\beta\|_1, \tag{175}$$

$$X \in \mathbb{R}^{286 \times 13237}, \quad \beta \in \mathbb{R}^{13237}, \quad y \in \mathbb{R}^{286},$$

where $\{\beta_g\}_{g=1}^G$ denote possibly overlapping groups of coefficients, and $w_g > 0$ are group-specific weights. We evaluate overlapping group LASSO on the GSE2034 breast cancer gene expression dataset, a widely used benchmark for high-dimensional, low-sample-size prediction in genomics. The dataset consists of primary tumor samples profiled using Affymetrix microarrays, with a binary clinical outcome indicating disease relapse. Gene expression values are mapped from probes to gene symbols using platform annotations, averaged across duplicate probes, log-transformed and standardized. Overlapping groups are constructed from curated KEGG pathways. Analytical approach is solved using FoGLASSO as recommended, (Yuan et al., 2011). DYS-HJ approach uses a backtracking scheme implemented by (Pedregosa & Gidel, 2018) with the HJ-Prox called on the overlapping group LASSO.

