# OpenReview forum: "Operator Splitting with Hamilton-Jacobi-based Proximals"
_ICML.cc/2026/Conference — ICML 2026 regular_

### Official Review · Reviewer_kRrA · 2026-03-09

**Soundness:** 3
**Presentation:** 3
**Significance:** 2
**Originality:** 2
**Overall Recommendation:** 4
**Confidence:** 3

**Summary:**

This paper studies how Hamilton–Jacobi-based proximal operators (HJ-Prox) can be integrated into classical operator-splitting optimization algorithms. The authors propose replacing exact proximal operators with a Monte-Carlo–based approximation derived from Hamilton–Jacobi PDE theory.
Empirical experiments on several convex optimization problems (e.g., LASSO, trend filtering, total variation denoising, multitask learning, and overlapping group LASSO) demonstrate that the proposed approach achieves solutions comparable to analytic proximal implementations while enabling the use of splitting algorithms when proximal operators are unavailable or expensive to compute.
The authors appear to present a major challenge: enabling operator-splitting methods to operate without closed-form proximal operators by using a stochastic approximation derived from Hamilton–Jacobi theory. The authors seek to consider a major question in modern optimization: whether proximal-based algorithms can remain effective when proximal operators are replaced by approximate zeroth-order estimators.

**Compliance With Llm Reviewing Policy:**

Affirmed.

**Final Justification:**

My concerns have been adequately addressed.

**Key Questions For Authors:**

1. The theory suggests increasing sample sizes across iterations, yet experiments use a fixed 𝑁. Why does this work empirically? Is there an informal explanation or heuristic theory?
2. How does HJ-Prox compare to classical approaches where the proximal subproblem is solved approximately using inner iterations?
3. What is the computational cost of evaluating HJ-Prox for very high-dimensional problems?
4. Could variance-reduction techniques or importance sampling improve the estimator?

**Limitations:**

The practical tuning of δ and N should be described in more detail.
Some notation related to the δ schedule could be clarified.
Figures focus heavily on visual similarity of solutions; quantitative metrics would strengthen the presentation.

**Strengths And Weaknesses:**

Strengths:
Operator splitting methods are widely used in machine learning and convex optimization. Their practical applicability is often limited by the requirement of computing closed-form proximal operators. Addressing this limitation is meaningful and well motivated.
The paper presents a coherent framework showing how HJ-Prox can be incorporated into multiple splitting methods by interpreting them as perturbed Krasnosel’skii–Mann iterations. This provides a systematic view of convergence for several algorithms within one analysis.
A key practical advantage is the ability to combine exact proximals and approximate proximals within a single splitting scheme. The hybrid strategy illustrated in experiments is compelling and aligns with realistic optimization pipelines.
Weaknesses:
The main conceptual ingredient—the Hamilton–Jacobi Monte Carlo approximation to proximal operators—comes from prior work (e.g., Osher et al.). The novelty of the present paper lies mainly in embedding this approximation inside splitting methods, rather than introducing a fundamentally new optimization paradigm.While the extension is useful, it may be perceived as incremental from a theoretical standpoint.
All experiments are classical convex problems. While appropriate for proximal theory, they are relatively simple from a modern machine learning perspective. Demonstrations on larger-scale ML systems or deep-learning-related optimization would increase impact.
The method relies on Monte Carlo sampling inside each proximal step. The paper does not provide detailed runtime comparisons or complexity analysis relative to standard solvers. For practitioners, runtime efficiency is critical.

---

> ### Author Rebuttal · Authors · 2026-03-30
>
> Thank you for taking the time to read our paper thoroughly. Below we address your comments.
>
> **"... novelty of the present paper lies mainly in embedding this approximation inside splitting methods,..."** While HJ-Prox exists in prior works, we re-emphasize part of our answer for another reviewer: the main technical contribution lies in Thm 3.3, where we develop a novel probabilistic bound for the proximal operator. This result constitutes the core of the analysis and, to the best of our knowledge, has not appeared in prior work on proximal methods. While propagating this result through the operator splitting framework follows more familiar lines, it is fundamentally enabled by Thm 3.3. Existing approaches typically assume access to exact integrals or expectations when evaluating the proximal operator. Thus, we do not believe that this is incremental from the theoretical standpoint. Thank you, we have clarified this in our draft.
>
> **On Expanding to Deep Learning/Non-Convex Problems**
> Our focus in this work is to introduce and rigorously analyze a new framework for integrating HJ-Prox into operator splitting methods, with particular emphasis on the substantial technical machinery required to establish Thm 3.3. To that end, we deliberately focus on convex problems, where proximal operators play a central role and comparisons to exact baselines are clean and controlled. But the reviewer raises an interesting point. There is existing work showing that HJ-Prox combined with proximal point methods performs well in nonconvex settings [1], and since the proof does not take into account the Monte-Carlo approximation, our framework may play a key role in extending these ideas to more general splitting methods. This is an interesting direction for future work. Thank you.
>
> **On Fixed N** This is a great point.  Fig 4 provides intuition for empirical results: under fixed $\delta$ and $N$, HJ-based methods converge to a solution neighborhood, visible as nonzero fixed-point residual floors. This is consistent with a persistent error regime in which exact convergence is replaced by neighborhood behavior. We refer to our response to Reviewer 1scs discussing the gap between empirical performance and theoretical requirements.
>
> **On Comparing HJ-Prox to Inner Iteration Problems with Inner Iterations** Two of our experiments directly address this: the overlapping group LASSO (Fig 5) and multitask learning (Fig 2). In Fig 5, the baseline requires FoGLASSO, a specialized dual solver [2]. In Fig 2, the baseline uses FISTA for the nuclear norm and Dykstra's algorithm for the mixed row/column group penalties.
> In both cases, the objective values remain close and the solutions are nearly identical up to numerical precision.
>
> **On Runtime Comparisons** We refer to our answer to Reviewer PKBA "On Runtime Comparison", where we now include an additional experiment comparing runtime of the HJ-Prox to an inner solver.
>
> **Cost of Evaluating HJ-Prox for High-Dimensional Problem** The cost is $N$ x $C_f$, where $C_f$ is the cost of a function evaluation, and $N$ is the number of samples. We also note that HJ-Prox is embarrasingly parallel, since each Monte Carlo sample can be evaluated independently (see response to Reviewer 1scs for a discussion on parallelism). With a simple function like L1 (and without using any tailored parallelism), this is how HJ-Prox scales:
>
> | Dimension   | N=100    | N=500     | N=1,000   |
> |-------------|----------|-----------|-----------|
> | 10,000      | 2.35 ms  | 5.80 ms   | 8.43 ms   |
> | 100,000     | 6.06 ms  | 18.42 ms  | 36.18 ms  |
> | 1,000,000   | 31.04 ms | 177.29 ms | 338.19 ms |
>
> **On Variance Reduction** This is an excellent point! We think importance sampling/variance reduction is precisely the future direction to make this method more efficient.
>
> **Tuning of $N$ and  $\delta$** In our experiments, N=1000 works well across all settings, with larger values yielding only marginal improvements. For tuning of delta please see "On details for choice of $\delta_k$ and $\alpha_k$" in response to PKBA. We have updated our Appendix H.
>
> **On Quantitative Metrics** We believe the paper already includes a range of quantitative evaluations, particularly in Fig 4 and 5. We would appreciate any specific suggestions on additional quantitative metrics by the reviewer.
>
> [1] Global Solutions to Nonconvex Problems by Evolution of Hamilton-Jacobi PDEs, Communications on Applied Mathematics and Computation, 2024
>
> [2] Efficient Methods for Overlapping Group Lasso, NeurIPS 2011

---

> > ### Author Rebuttal · Reviewer_kRrA · 2026-04-02
> >
> > Thank you to the authors for the rebuttal. While I appreciate the clarifications provided, my main concerns have not been sufficiently addressed.

---

### Official Review · Reviewer_PKBA · 2026-03-11

**Soundness:** 3
**Presentation:** 3
**Significance:** 2
**Originality:** 2
**Overall Recommendation:** 4
**Confidence:** 3

**Summary:**

This paper studies operator splitting algorithms in which proximal operators are approximated by HJ-Prox, a Hamilton-Jacobi-based derivative-free Monte Carlo method. It establishes convergence results for several existing splitting methods, including PPM, DRS, DYS, and PDHG. Numerical experiments show that these methods equipped with HJ-Prox perform well, even for problems whose proximal operators are difficult or unavailable in closed form.

**Compliance With Llm Reviewing Policy:**

Affirmed.

**Final Justification:**

It requires some revisions, but I found it theoretically sound and had a favorable impression of the numerical results.

**Key Questions For Authors:**

1. In the proofs of Theorems 3.5, 3.8, and 3.10, Theorem 3.3 is invoked to control the HJ-Prox approximation error. However, Theorem 3.3 assumes that the underlying function is convex, lower semicontinuous, and $L$-Lipschitz, whereas the statements of Theorems 3.5, 3.8, and 3.10 do not appear to assume the corresponding Lipschitz continuity of $f+g$, $f,g$, and $g^*,f$, respectively. Could the authors clarify why Theorem 3.3 is applicable in these settings? Also, the notation using the same symbol $L$ for both $L$-Lipschitz continuity and $L$-smoothness is somewhat confusing, since these are different notions. It would improve readability to distinguish these constants explicitly.

2. Could the authors provide more details on the choice of $\delta_k$ used in the numerical experiments, including the specific schedule and the value of $p$? It would also be helpful to know whether other schedules satisfying the assumptions were tested, and how sensitive the practical performance of HJ-Prox is to the choice of $p$.

3. In Theorem 3.6, the step-size condition is stated as $0 < t_k < 2/L$, while the example immediately afterwards uses $t_k = 1/k$. Could the authors clarify how this choice is intended to satisfy the stated condition? More generally, in Appendix D, some parts of the argument appear to use a stronger condition such as $t_k \leq 1/L$. It would be helpful if the authors could clarify the precise step-size requirement. In addition, Assumption 3.7 fixes $t$ across iterations. Is this restriction essential to the method itself, or is it only needed for the current convergence analysis?

**Limitations:**

yes

**Strengths And Weaknesses:**

***Strengths***

- The paper studies an interesting and practically meaningful direction by integrating HJ-Prox into existing operator splitting algorithms for optimization problems in which proximal operators are difficult or unavailable in closed form.

- The paper provides convergence analysis for PPM, PGD, DRS, DYS, and PDHG with HJ-Prox, which makes the work more substantial from a theoretical perspective.

- The numerical experiments are generally convincing. In problems where closed-form proximal operators are available, the proposed methods perform comparably to the corresponding exact algorithms. In more challenging settings, the method still produces reasonable and competitive solutions, even if the convergence is slower.

***Weaknesses***

1. The paper provides concrete examples of parameter schedules for $\delta_k$ and $\alpha_k$. It may nevertheless be helpful to include a brief discussion of how the decay exponent $p$ should be chosen in practice.

2. Although the paper establishes asymptotic convergence, it offers little insight into convergence rates or iteration complexity. This limits the significance of the theoretical results, since it remains unclear how the use of HJ-Prox affects the decrease of the objective value or a stationarity measure, as well as the overall computational cost.

3. The experimental section does not report runtime comparisons. Since HJ-Prox may have a significantly different per-iteration cost from exact proximal updates, comparing methods only in terms of iteration count is not fully sufficient for evaluating practical efficiency.

---

> ### Author Rebuttal · Authors · 2026-03-30
>
> Thank you for  your thoughtful feedback. We especially appreciate the acknowledgment of the usefulness of this work for problems with unavailable/difficult proximals.
>
> **On Assuming L-Lipschitz, Convexity, Proper, and LSC** Thanks for pointing this out. We mistakenly did not include this in Thm 3.5, 3.8, and 3.10. We have included this in our revised draft. We also clarified the different Lipschitz constants by using different variables names.
>
> **On Insight into Convergence Rate** This is a great point. Since our analysis reduces to controlling the error in a perturbed KM iteration, we suspect the convergence rate should mirror that of the underlying KM iteration if the error is appropriately controlled. This can be seen in all of our experiments. We included remarks about this in our draft.
>
> **On Runtime Comparison** We have now included runtime comparisons in our appendix section for the proximal evaluation of the overlapping group Lasso penalty, which we also show here. In particular, we compare the runtime of FoGLasso's specialized inner solver (used in Fig 5) against HJ-Prox. HJ-Prox uses N=1000 and FoGLasso is implemented as recommended. HJ-Prox achieves better run-time as dimensionality scales up. We consistently achieve relative error within $10.5$% of FoGLASSO's solution.
>
> | Dimension | FoG Time | HJ-Prox Time |
> |-----------|----------|---------------|
> | 50,000    | 358.9 ms | 401.3 ms      |
> | 100,000   | 1.01 s   | 739.4 ms      |
> | 150,000   | 1.55 s   | 1.07 s        |
> | 200,000   | 2.06 s   | 1.50 s        |
> | 250,000   | 2.47 s   | 1.83 s        |
>
> This shows that HJ-Prox can be competitive for problems with functions that do not admit closed form proximals. We remark that runtime comparisons depend highly on the numerical implementation and is therefore not the focus of this work. However, it is worth noting that since HJ-Prox is embarrassingly parallel due to independent function evaluations, it has the potential be faster. Experiments were run on a single Apple M2 Pro chip 16GB.
>
> **On details for choice of $\delta_k$ and $\alpha_k$**
> - Choice of $\delta_k$ (and p): For the specific figures in the paper, we used $\frac{1}{k^{2.00001}}$. In all of our experiments, we find that $\delta_k = \frac{1}{k^{2 + p}}$ for $p \approx 0$ works best (as shown in table below on the Sparse Lasso Group objective function). Intuitively, an aggressive $\delta$ schedule causes the sampling variance (in eq (2)) to shrink too rapidly and we won't explore far away from $x_k$. We even observe $p = 0$ works well empirically.
> | Rate           | Final Objective Value |
> |----------------|----------------------|
> | 1/k            | 457.22               |
> | 1/k²           | 447.47               |
> | 1/$k^{2.00001}$  | 447.39               |
> | 1/k³           | 508.63               |
> | 1/k⁴           | 526.74               |
> | Baseline       | 447.04               |
> - Choice of $\alpha_k$: Making $\alpha_k$ decay too fast also increases the theoretically sufficient sample size through  $\frac{1}{\alpha_k}$. A subtle but important point is that $\alpha_k$ is a proof-level confidence schedule, not an algorithmic hyperparameter, we have edited the paper for clarity. The more favorable power-law choice is therefore the slower summable ones that still decays fast enough to well control the errors.
>
> We have incorporated feedback and included information regarding practical selection of schedules into our Appendix H.
>
> **On Step-size condition, Appendix D and Assumption 3.7** Thank you for catching this. There is indeed a mismatch: the proof in Appendix D (Lemma D.1) requires $t_k \leq 1/L$ for fejer monotonicity, which is stronger than the $0 < t_k < 2/L$ stated in Thm 3.6. We have revised Thm 3.6 to adopt the tighter condition $0 < t_k \leq 1/L$ for consistency. The example $t_k = \gamma/k$ with $\gamma \leq 1/L$ satisfies this for all $k$. Regarding Assumption 3.7: fixing $t$ across iterations is required by the KM framework to ensure the solution set $S = Fix(T)$ remains stable. Extending DRS, DYS, and PDHG to time-varying step sizes would require a generalization of the operator theory machinery beyond the current framework. Thanks for pointing these out!

---

> > ### Author Rebuttal · Reviewer_PKBA · 2026-04-01
> >
> > Thanks for your rebuttal. I remain my original score.

---

### Official Review · Reviewer_1scs · 2026-03-13

**Soundness:** 4
**Presentation:** 4
**Significance:** 4
**Originality:** 4
**Overall Recommendation:** 5
**Confidence:** 4

**Summary:**

Summary: The author’s main theorems (3.5, 3.6, 3.8, 3.9, and 3.10) establish that one may replace exact proximal operators with the zeroth order Hamilton-Jacobi proximal (HJ-Prox) Monte Carlo approximation in the Proximal Point Method (PPM), Proximal Gradient Descent (PGD), Douglas-Rachford Splitting (DRS), Davis Yin-Splitting (DYS), and Primal-Dual Hybrid gradient (PDHG) algorithms and in particular that it preserves the a.s. convergence of these algorithms to a minimizer of the objective function f+g in standard composite optimization. The authors also do some numerical experiments to empirically validate their theoretical results.

**Compliance With Llm Reviewing Policy:**

Affirmed.

**Final Justification:**

This paper is novel and interesting, and the theoretical results provide a convincing starting point for replacing proximal operators with the zeroth order HJ Monte Carlo approximation which has nice connections to classical HJ PDE theory. As the paper is very clear and well-written, I recommend accept.

**Key Questions For Authors:**

-How much do the authors think that the sample complexity and parameter settings needed for the KM iterate condition can be improved?

-How do the authors explain that the finite sample size of $N=1000$ works well empirically, even if in theory this violates Borel-Cantelli summability/the KM iteration condition? Is T3.3 too pessimistic, and do the authors think the conservatism required in the assumptions can be significantly relaxed with a different proof approach?

-The authors note initially that the HJ-prox holds for weakly-convex functions f (Ryu & Yin 2022), but then prove the results for convex f. Why did they weaken this condition/ what are the difficulties showing their results for weakly-convex functions?

-It’s worth noting that the HJ-Prox requires sampling $N$ points for each iteration for the Monte-Carlo estimate of the proximal operator. Thus, it would be worth mentioning in the paper (e.g. in a limitations section) the time-complexity trade-off of using this Monte-carlo estimate of the prox operator.

**Limitations:**

Yes

**Strengths And Weaknesses:**

Strengths:

-The paper is well written and clear, and the proof technique is very simple. I didn’t read the appendix in detail, but the proof techniques are clear from the main text. In particular, the authors use the classical convergence of KM iterates in T3.1 which states that convergence to a fixed point holds for an averaged operator T_k if the error sequence is strictly summable and the operator satisfies appropriate closedness conditions. The error of the HJ-Prox step separates into a deterministic error bound for HJ-Prox [which was classically proved in the context of classic PDE theory for viscosity solutions (i.e. Laplacian regularized HJ) to the HJ PDE] and a stochastic Monte Carlo variance error bound. The former can be handled by appropriate setting of \delta_k for summability, and the a.s. convergence of the latter is handled (to match the summability being deterministic in T3.1) by use of Borel Cantelli and appropriate restriction on the smoothness, sample complexity, proximal step, and the tail probability bound parameters. Then, the authors essentially show that for the PPM, etc schemes that the operator satisfies the KM requirements for the conclusion that the HJ-Prox splitting scheme yields a sequence (x_k) converging a.s. to the minimizer.

Weaknesses:

-To actually invoke Borel-Cantelli and a.s. convergence is perhaps much too strong in practice, and requires sub-exponential/exponential sample-complexity. The experiments indicate that a fixed Monte Carlo sample size of 1000 works well, which suggests a gap between the empirical performance and the theoretical requirements for the proof to go through. Thus, I am concerned that the theoretical results are somewhat too weak/restrictive, and that there remains a significant gap from the formal KM summability requirements. Another thing worth noting is the global L-Lipschitz requirement needed says that $J_*$ has exponential dependence on $L^2$, perhaps the requirement is restrictive and could perhaps be relaxed to improve the bound.

-There are a few typos/broken links (e.g. line 847)

---

> ### Author Rebuttal · Authors · 2026-03-30
>
> Thanks you for the positive and constructive feedback! We address your comments below.
>
> **On Gap Between Empirical Performance and Theoretical Requirements (and thoughts on improving sample complexity)** We agree that almost sure convergence is a strong requirement in practice, and that it is likely sufficient for the Monte Carlo errors to be controlled in expectation (or with high probability), rather than being summable almost surely. A main culprit from Borel Cantelli is the summability of $\sum \alpha_k$, which drives the sample complexity upward for the overall errors to remain summable under the perturbed operator framework. This suggests the possibility of deriving more practical results under weaker probabilistic notions. The same can be said about the global L-Lipschitz assumption. We suspect it can be relaxed to a local Lipschitz requirement.
> That said, we view our current analysis as providing a clean and broadly applicable baseline guarantee, upon which more refined results can be developed. We think the sample parameter complexity can be significantly improved, so that even with fixed sample size we can converge to a neighborhood around the minimizer, as is the case in traditional SGD-type convergence. This may explain why we empirically observe stable convergence even with a fixed sample size (e.g., N=1000), despite not satisfying the summability conditions required by the theory. Fig 4 demonstrates this directly: under fixed $\delta$ and $N$ HJ-based methods converge to a small neighborhood, visible as nonzero fixed-point residual floors. When $\delta_k$ is decreased with fixed $N$, this neighborhood shrinks over the iteration horizon. We are actively exploring these directions.
>
> **On Time Complexity Discussion** The reviewer raises a valid point that HJ-Prox requires $N$ points for the Monte Carlo estimate. We want to point out that this can actually be a computational strength. Unlike iterative inner solvers, HJ-Prox computation consists of $N$ independent function evaluations, which makes it embarrassingly parallel. The wall-clock cost of an HJ-Prox evaluation distributed throughout nodes may be lower than the sequential inner-loop alternatives. We provide additional detail and run-time experiments displayed in responses to PKBA and kRrA. We have included discussion regarding time-complexity trade off and parallel computation in the paper.
>
> **On Weakly-Convex f** This is a great observation. Indeed, we assume convexity of f, even when the original HJ-Prox only requires weak-convexity. We needed this for technical reasons (proof of Thm 3.2 and Thm 3.3). But to be consistent with the discussion above, we think this assumption can be relaxed to weak convexity of f if we weaken the convergence guarantee from a.s. to convergence in expectation.
>
> **On Typos** Typos have been fixed, thanks!

---

> > ### Author Rebuttal · Reviewer_1scs · 2026-03-31
> >
> > All of my questions have been resolved. I maintain my score of Accept.

---

### Official Review · Reviewer_6FHu · 2026-03-15

**Soundness:** 4
**Presentation:** 4
**Significance:** 4
**Originality:** 3
**Overall Recommendation:** 4
**Confidence:** 5

**Summary:**

This paper studies the use of Hamilton-Jacobi-based proximal approximations (HJ-Prox) within classical operator splitting algorithms. Building on the Osher et al. 2023 prior work that introduced HJ-Prox as a Monte Carlo approximation to proximal operators, the authors develop a unified framework showing that replacing exact proximal steps with HJ-Prox approximations preserves convergence guarantees for several standard methods, including proximal gradient descent, Douglas–Rachford splitting, Davis–Yin splitting, and PDHG. The analysis interprets these algorithms as perturbed fixed-point iterations and derives conditions under which the approximation errors remain summable, ensuring convergence. The paper is clearly written, and the theoretical development is systematic.

**Compliance With Llm Reviewing Policy:**

Affirmed.

**Key Questions For Authors:**

.

**Limitations:**

.

**Strengths And Weaknesses:**

Strength of the paper is that it presents a clean theoretical framework for analyzing operator-splitting algorithms with approximate proximal operators. The convergence analysis is technically sound and leverages classical tools from fixed-point theory in a clear way. The presentation is also well organized, and the paper provides several illustrative experiments showing that HJ-Prox can act as a drop-in replacement for proximal operators in a variety of standard optimization problems.

At the same time, the conceptual novelty of the work is somewhat limited relative to existing literature. The HJ-Prox approximation comes from the prior work, and the present paper primarily presents a convergence analysis for inserting this approximation into standard splitting methods. The techniques used in the analysis rely on standard techniques from operator and fixed point theory.

The experiments could also be strengthened. Many of the presented experiments focus on problems whose proximal operators are already available in closed form (e.g., LASSO or total variation), so the results mainly demonstrate that the proposed method reproduces solutions obtained by existing approaches rather than solving new classes of problems. Demonstrating clear advantages in settings where proximal operators are genuinely difficult to compute would strengthen the overall impact of the work.

---

> ### Author Rebuttal · Authors · 2026-03-30
>
> Thank you for the constructive feedback!
>
> **On Conceptual Novelty**
> We emphasize that our analysis departs significantly from standard fixed-point theory. In particular, the main technical contribution lies in Thm 3.3, with the novel probabilistic bound for the proximal operator. This result constitutes the core of the analysis and, to the best of our knowledge, has not appeared in prior work on proximal methods. While propagating this result through the operator splitting framework follows more familiar lines, it is fundamentally enabled by Thm 3.3. Existing approaches typically assume access to exact integrals or expectations when evaluating the proximal operator. In contrast, our analysis explicitly accounts for the use of samples in practice. Thus, we believe this is a meaningful step toward closing the gap between theoretical guarantees and practical implementations. We appreciate this comment, and we intend to make this more explicit in our draft.
>
> **On Problems Whose Proximal Operators Not Available**
> As this work is intended as a proof of concept for integrating HJ-Prox into operator splitting methods, we believe it is important to first validate the approach on problems where the proximal operator is analytically known. This allows for a clear and controlled comparison.
> That said, we respectfully disagree with the claim that there are not enough relevant problems without closed-form proximal operators. Several of the examples we consider (Sections 4.2–4.4), feature proximals that are mathematically challenging to compute, which we list below:
>
> - In trend filtering, the proximal operator is not available in closed form, and standard approaches rely on product-space reformulations together with ill-conditioned linear solves.
> - In multitask learning, analytical baselines require iterative SVD-based inner routines combined with Dykstra- or FISTA-type procedures.
> - Similarly, in overlapping group lasso, the benchmark FoGLASSO method relies on a dual reformulation and specialized solvers.
>
> This comment highlights an important aspect of our contribution, and we have revised the manuscript to make this point more explicit.

---

> > ### Author Rebuttal · Reviewer_6FHu · 2026-04-06
> >
> > Thank you for the response. I am happy with the paper, and I will maintain my score.

---

### Decision · Program_Chairs · 2026-04-30

**Decision:**

Accept (regular)

**Comment:**

This paper studies classical operator splitting algorithms in which the proximal operator is replaced by Hamilton-Jacobi-based proximal approximations (HJ-Prox).

The writing is clear and clean, the theory is solid, and the experiments support the soundness of the framework. However, I agree with the reviewers that it is not yet clear how efficiently the approach can be used in broader settings beyond the standard cases considered in the paper. On the negative side, the paper does not bring a major conceptual surprise, and instead combines existing ingredients in a somewhat expected way.

Overall, the contribution is solid. The reviewers unanimously recommend acceptance, and I concur.